

# Fluorine NMR study of proline-rich sequences using fluoroprolines

Davy Sinnaeve[1,2], Abir Ben Bouzayene[3], Emile Ottoy[4], Gert-Jan Hofman[4,5], Eva Erdmann[3],

Bruno Linclau[5], Ilya Kuprov[5], José C. Martins[4], Vladimir Torbeev[6] and Bruno Kieffer[3]

[1]Univ. Lille, Inserm, Institut Pasteur de Lille, CHU Lille, U1167 – Labex DISTALZ – RID-AGE – Risk Factors and Molecular Determinants of Aging-Related Diseases, F-59000 Lille, France

[2]CNRS, ERL9002 - Integrative Structural Biology, F-59000 Lille, France

[3]Departement of Integrative Structural Biology, IGBMC, Université de Strasbourg,
INSERM U1258, CNRS UMR7104, 1, rue Laurent Fries, F-67404 Illkirch, France

[4]Department of Organic and Macromolecular Chemistry, Ghent University,
Campus Sterre, S4, Krijgslaan 281, B-9000 Gent, Belgium

[5]School of Chemistry, University of Southampton, Southampton SO17 1BJ, United Kingdom

[6]Institut de Science et d'Ingénierie Supramoléculaires (ISIS), International Center
for Frontier Research in Chemistry (icFRC), University of Strasbourg,

CNRS UMR 7006, F-67000 Strasbourg, France

*Correspondence to*: B. Kieffer (kieffer@igbmc.fr) and D. Sinnaeve (davy.sinnaeve@univ-lille.fr)

**Abstract.** Proline homopolymer motifs are found in many proteins; their peculiar conformational and dynamic properties are often directly involved in those proteins' functions. However, the dynamics of proline homopolymers is hard to study by

NMR due to lack of amide protons and small chemical shift dispersion. Exploiting the spectroscopic properties of fluorinated prolines opens interesting perspectives to address these issues. Fluorinated prolines are already widely used in protein structure engineering – they introduce conformational and dynamical biases – but their use as [19]F NMR reporters of proline conformation has not yet been explored. In this work, we look at model peptides where Cγ-fluorinated prolines with opposite configurations of the chiral Cγ centre have been introduced at two postions in distinct polyproline segments. By looking at

the effects of swapping these (4*R*)- and (4*S*)-4-fluoroprolines witin the polyproline segments, we were able to separate the intrinsic conformational properties of the polyproline sequence from the conformational alterations instilled by fluorination. We assess the fluoroproline [19]F relaxation properties, and exploit the latter in elucidating binding kinetics to the SH3 domain.






## 1. Introduction

The use of $^{19}$F nuclei in medical and biological magnetic resonance is gaining popularity (Zhang et al., 2017). Since the pioneering incorporation of *p*-fluorophenylalanine (Chaiken et al.,1973) into ribonuclease-S' analogues, dozens of $^{19}$F-labelled amino acid analogues have been evaluated (Odar et al., 2015; Mei et al., 2020; Muttenthaler et al.; 2021, Salwiczek

et al., 2012). Common ways to incorporate fluorinated amino acids in peptides or proteins are: (a) solid phase chemical synthesis (Behrendt et al., 2016); (b) post-translational addition of fluoroalkyl groups to reactive amino acid side chains (Liu et al., 2012); (c) addition of fluorinated precursors, such as fluoroindole, to bacterial culture media prior to protein overexpression (Crowley et al., 2012); (d) using recombinantly expressed orthogonal amber codon tRNA/tRNA synthetase pairs (Sharaf et al., 2015; Gimenez et al., 2021; Gee et al., 2016; Kitevski-LeBlanc et al., 2012). The advantages of $^{19}$F nuclei in

biological NMR are lack of background signals, high magnetogyric ratio, 100% natural abundance, and the sensitivity of $^{19}$F chemical shift to the chemical environment (Rastinejad  et al., 1995). Fluorine chemical shift range (~50 times wider than that of $^{1}$H) makes it possible to study faster chemical exchange processes than those accessible to $^{1}$H- and $^{13}$C-based methods. This is useful in biomolecular interaction studies and examples include the deciphering of the signal transduction pathways through the β2-adrenergic trans-membrane receptor (Liu et al., 2012), the study of conformer interconversion and

allostery that drive the catalytic process in the bacterial enzyme fluoroacetate dehalogenase ( Kim et al., 2017), the monitoring of both kinetic and equilibrium thermodynamic binding parameters of a fluorine-labeled Src homology 3 (SH3) protein domain to peptides containing proline-rich motifs (PRM) (Stadmiller et al., 2020), and the folding study of a small protein domain (Evanics et al., 2007). A downside of $^{19}$F is high chemical shift anisotropy (CSA) – particularly in aromatic rings – resulting in rapid transverse relaxation and broad lines for large biomolecules at high magnetic fields (Kitevski-LeBlanc et

al., 2012), although the recently proposed $^{19}$F-$^{13}$C aromatic TROSY experiment has alleviated this to some extent (Boeszoermenyi et al., 2019).

Fluorination is well known for its significant impact on the properties of organic molecules (Aufiero et al., 2018; Gillis et al., 2015, Berger et al., 2017). Apart from altering the interaction with the solvent (*cf.* hydrophobicity), replacing a hydrogen with fluorine can produce significant structural changes. Firstly, the volume of the moiety increases. Although fluorine is

often considered isosteric to hydrogen based on its similar Van der Waals radius ($r_{VdW}$(F) = 1.47 Å *vs.* $r_{VdW}$(H) = 1.20 Å) (Bondi et al., 1964) its covalent radius is significantly larger ($r_{cov}$(F) = 0.57 Å *vs.* $r_{cov}$(H) = 0.31 Å) due to greater C–F bond length (Cordero et al., 2008; O'hagan et al., 2008). As a result, fluorine may perturb the protein fold when the fluorinated side-chain is tightly packed within a protein structure. Secondly, the polar C–F bond brings in additional charge and polaris-ability effects (Salwiczek et al., 2012). In aromatic side chains, swapping a single hydrogen for fluorine does not normally

(there are exceptions (Salwiczek et al., 2012; Boeszoermenyi et al., 2020; Yoshida et al., 1960)) alter the fold or the function of the protein (Welte et al., 2020). In contrast, fluorinating an aliphatic CH group can radically change local rotamer populations (O'hagan et al., 2008; O'Hagan et al., 2012). This effect has been put to good use (Salwiczek et al., 2012; Berger et al., 2017), particularly in fluorinated prolines (Kubyshkin et al., 2021; Verhoork et al., 2018; Newberry et al., 2016).

Proline is the only proteinogenic amino acid with a secondary amino group, thus allowing for the *cis*-peptide bond isomer to

be significantly populated. In addition, its pyrrolidine ring can adopt either a Cγ-*endo* or Cγ-*exo* conformation, with a slight preference for the former. Single or double fluorination at the β- and/or γ-positions shifts these conformational equilibria in a stereospecific way. For instance, (4*R*)-fluorination favours the Cγ-*exo* ring conformer and enhances the *trans* isomer population, while (4*S*)-fluorination does the opposite (Fig. 1) (Eberhardt et al., 1996; Panasik et al., 1994).





This is caused by stabilizing $^{C-H}\sigma_{(HOMO)} \rightarrow {}^{C-F}\sigma^*_{(LUMO)}$ hyperconjugative delocalization, which is maximal when the C–H

bond is antiperiplanar to the C–F bond, a phenomenon generally known as *gauche* effect (Thiehoff et al., 2017). The increased amount of Cγ-*exo* conformer in (4R)-FPro in turn increases the *trans* isomer population since this is the most favourable configuration for further stabilizing hyperconjugative n→π* delocalization between carbonyl groups in successive peptide bond (Newberry et al., 2016). Similarly, the reduced Cγ-*exo* population as well as the steric impact from the longer C–F bond increases the population of the *cis*-isomer in (4S)-FPro. The increase or decrease of n→π* hyperconjugation have been

used to explain stabilization or destabilization of the polyproline-II (PPII) conformation in all-(4R)-fluorinated and all-(4S)-fluorinated oligoprolines respectively (Horng et al., 2006). The ability to control the conformational preference of individual proline residues is central to elucidating the role of proline conformation on the stability, folding, and aggregation of various proteins, such as collagen (Holmgren et al., 1998; Shoulders et al., 2009), β2-microglobulin (Torbeev et al., 2015; Torbeev et al., 2013) and tau. (Jiji et al., 2016)

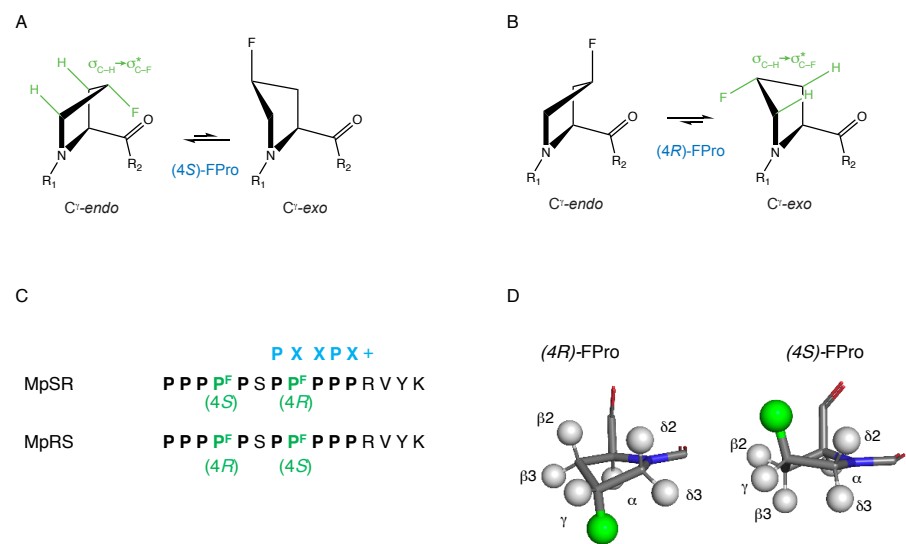


**Figure 1:** Cγ-*endo* and Cγ-*exo* puckering of the pyrrolidine ring in (4S)- and (4R)-fluoroprolines shown respectively in **A** and **B**. The *gauche effect* stabilizes the Cγ-*endo* conformer of (4S)-fluoroproline, whereas the Cγ-*exo* conformer is favoured in (4R)-fluoroproline. **C**: Fluoroprolines incorporated in a proline-rich sequence at two positions 4 and 8 highlighted in

green. Two peptides are studied: in MpSR (4S)-fluoroproline is inserted at the fourth position and (4R)-fluoroproline is inserted at the eighth position. In MpRS, the positions of the (4R)- and (4S)-fluoroprolines are reversed, placing them in the position fourth and eighth, respectively. The canonical SH3 domain binding motif is shown in blue. **D, left side**: 3D model of (4R)-fluoroproline where Hγ3 is substituted by a fluorine atom. The carbonyl group and the fluorine atom point towards opposite sides. **D, right side**: 3D model of (4S)-fluoroproline where Hγ2 is substituted by a fluorine atom. The carbonyl

group and the fluorine atom point towards the same direction.



Surprisingly, despite the well-established use of FPro residues in chemical biology, they have so far not found any application as [19]F NMR reporters in protein studies, in contrast to aromatic amino acids. Yet their potential is clear in this regard, given the abundance of proline in intrinsically disordered protein sequences, the prominent role of proline-rich regions as

sites for protein-protein interaction and post-translational modification, the relatively small CSA of [19]F nuclei in prolines (thus, narrow lines), and the challenge of detecting minor *cis* isomers (Theillet et al., 2013). Possible explanations are the unknown [19]F NMR properties of these residues, as well as their undesirably strong conformational impact.

With the purpose of filling this gap, we have studied the impact of (4*R*)- and (4*S*)-FPro residues on the structure and dynamics of a polyproline peptide harbouring an SH3 binding motif, and used [19]F NMR to investigate the impact of fluorination on

the binding affinity. We designed model peptides containing (4*R*) and (4*S*) fluorinated prolines with a sequence based on the motif located at the C-terminal part of the retinoic acid hormone nuclear receptor RARγ that specifically binds to the third SH3 domain of the Vinexin β protein (Lalevee et al., 2010). First, we explored the impact of FPro introduction on the surrounding peptide sequence, and verified the preferred FPro ring pucker within the polyproline context. Next, we used [19]F relaxation analysis to gain insights into the local dynamics of the peptide. Finally, we monitored the interaction of the model

peptides with the vinexin β SH3 domain using [19]F NMR, and demonstrated that FPro conformational bias can be used to modulate the kinetics of protein binding to proline-rich motifs. This work paves the way to using fluoroprolines as [19]F NMR reporters in protein interaction studies, where the conformational bias caused by fluorine is exploited to obtain information on binding kinetics.

## 2. Results

### 110   2.1 Assignment and spectral analysis of model peptides

The model peptide sequences shown in Fig. 1C contain two segments of five prolines separated by a single serine, and terminate with a four-residue sequence (RVYK) required for the SH3 class II binding specificity. FPro residues were inserted at positions 4 and 8, which are not directly involved at the protein-peptide interface according to homology models of PP-II helices-SH3 complexes (Saksela et al., 2012). Position 4, located in the first polyproline segment, falls outside the expected

PXXPX+ binding motif while proline 8, which is located within the canonical SH3-PPII binding motif, is expected to be solvent-exposed. Thus, the fluorine atoms are not expected to contribute significantly to the protein-peptide binding interface. Two peptides were considered, with (4*R*)- and (4*S*)-FPro substitutions at positions 4 and 8 (hereafter named MpRS), or introduced at positions 8 and 4 (MpSR).

Full [1]H and [13]C chemical shift assignments of the non-proline and FPro residues in $D_2O$ were achieved using standard [1]H-[1]H

NOESY, [1]H-[1]H TOCSY and [1]H-[13]C HSQC experiments. The 8 non-fluorinated proline residues have very similar chemical shifts, but full assignment could still be achieved using a 2D [1]H-[13]C HSQC-NOESY experiment with very high [13]C digital resolution (*ca.* 4 Hz, see experimental section) (Fig. 2). For this, the spectral window was set to a narrow [13]C chemical shift region of 3 ppm containing the proline Cδ resonances. To avoid interference from folded [13]C-Hα autocorrelation peaks, a gradient-enhanced frequency-selective [13]C 180° refocusing pulse was applied in the HSQC experiment. At this spectral reso-

lution, the minute Cδ chemical shift dispersion (0.3 ppm for prolines 2 to 11 in MpSR) allowed resolving the sequential Hδ(*i*-1) to Hα(*i*) NOE cross-peaks and thus completing [1]H and [13]C chemical shift assignment of both peptides (Table 1).



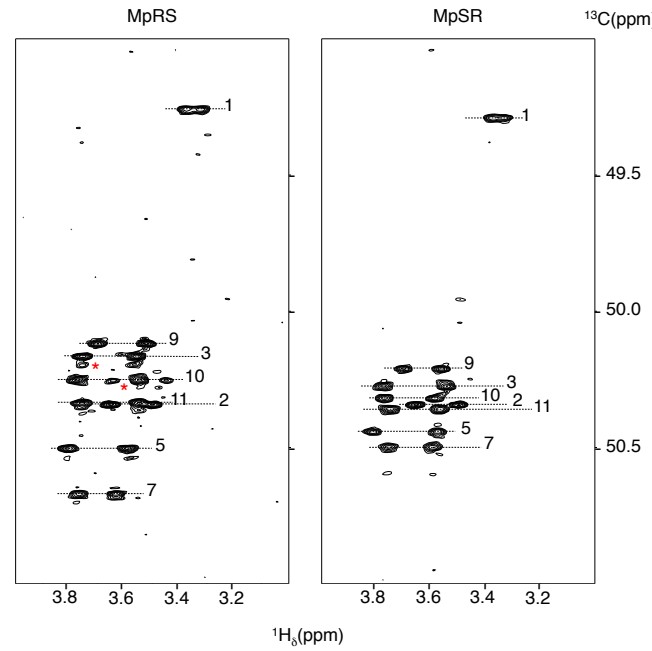

**Figure 2**: $^1$H-$^{13}$C HSQC-NOESY (mixing time: 80 ms) with a narrow window showing only the $^{13}$Cδ resonance region of both MpRS and MpSR peptides, recorded at 298 K and 700 MHz. The numbers indicate the position of the residue in the sequence. The red asterisks highlight minor forms of prolines.





**Table 1**: Chemical shift assignments of peptide MpSR and MpRS peptides. $\Delta\delta$ is the chemical shift difference between $^{13}C\beta$ and $^{13}C\gamma$ resonances used as indicator for *cis* and *trans* conformations of the Xaa-Pro peptide bond. $\Delta C\alpha$ is the chemical shift difference between the measured $^{13}C\alpha$ and the corresponding random coil values. Chemical shifts were measured in $D_2O$ at pH 7, 298 K and referenced to DSS-$d_6$.

**MpRS**

| | Hα | Hβ | Hγ | Hδ | Cα | Cβ | Cγ | Cδ | Δδ | ΔCα |
|---|---|---|---|---|---|---|---|---|---|---|
| Pro1 | 4.61 | 2.54 2.04 | 2.04 | 3.42 3.37 | 61.76 | 30.86 | 26.51 | 49.28 | 4.35 | -1.58 |
| Pro2 | 4.76 | 2.39 1.88 | 2.01 | 3.69 3.54 | 61.62 | 30.57 | 27.26 | 50.37 | 3.31 | -1.72 |
| Pro3 | 4.71 | 2.29 1.84 | 2.05 | 3.80 3.60 | 61.24 | 30.49 | 27.32 | 50.20 | 3.17 | -2.10 |
| (4R)-FPro | 4.89 | 2.73 2.03 | 5.44 | 4.20 3.81 | 59.72 | 37.22 | 95.66 | 56.52 | -58.44 | -3.62 |
| Pro5 | 4.45 | 2.28 1.88 | 2.01 | 3.85 3.64 | 62.86 | 32.06 | 27.26 | 50.51 | 4.80 | -0.48 |
| Ser6 | 4.70 | 3.85 3.72 | | | 56.46 | 62.97 | | | | -2.25 |
| Pro7 | 4.62 | 2.37 1.96 | 2.05 | 3.82 3.68 | 61.53 | 30.78 | 27.32 | 50.69 | 3.46 | -1.81 |
| (4S)-FPro | 4.86 | 2.62 2.38 | 5.41 | 4.07 3.96 | 60.35 | 36.90 | 95.63 | 56.50 | -58.73 | -2.99 |
| Pro9 | 4.69 | 2.3 1.87 | 2.02 | 3.74 3.56 | 61.22 | 30.52 | 27.25 | 50.13 | 3.27 | -2.12 |
| Pro10 | 4.68 | 2.31 1.87 | 2.02 | 3.82 3.59 | 61.19 | 30.73 | 27.25 | 50.25 | 3.48 | -2.15 |
| Pro11 | 4.37 | 2.26 1.83 | 2.00 1.98 | 3.8 3.59 | 62.82 | 32.01 | 27.26 | 50.33 | 4.75 | -0.52 |
| Arg12 | 4.24 | 1.68 1.67 | 1.54 1.47 | 3.13 3.13 | 55.80 | 30.80 | 26.96 | 43.17 | | -0.98 |
| Val13 | 4.06 | 1.94 | 0.85 0.83 | | 61.83 | 33.03 | 20.97 20.44 | | | -0.71 |
| Tyr14 | 4.57 | 3.01 2.87 | | 7.10 Hε6.78 | 57.51 | 39.09 | | 53.21 Cε37.98 | | -0.67 |
| Lys15 | 4.28 | 1.84 1.72 | 1.34 1.33 | 1.65 1.62 Hε2.92 | 55.16 | 32.72 | 24.63 | 28.84 Cε41.82 | | -1.80 |



**MpSR**

| | Hα | Hβ | Hγ | Hδ | Cα | Cβ | Cγ | Cδ | Δδ | ΔCα |
|---|---|---|---|---|---|---|---|---|---|---|
| Pro1 | 4.61 | 2.54 2.39 | 2.05 | 3.42 3.37 | 61.75 | 30.91 | 26.52 | 49.3 | 4.39 | -1.59 |
| Pro2 | 4.76 | 2.39 1.88 | 2.01 | 3.70 3.54 | 61.65 | 30.64 | 27.27 | 50.36 | 3.37 | -1.69 |
| Pro3 | 4.63 | 2.38 1.96 | 2.06 | 3.82 3.59 | 61.31 | 30.63 | 27.30 | 50.28 | 3.33 | -2.03 |
| (4S)-FPro | 4.87 | 2.59 2.43 | 5.42 | 4.07 3.99 | 60.32 | 36.97 | 95.09 | 56.55 | -58.12 | -3.02 |
| Pro5 | 4.43 | 2.27 1.88 | 2.02 | 3.85 3.62 | 63.06 | 31.89 | 27.27 | 50.44 | 4.62 | -0.28 |
| Ser6 | 4.71 | 3.84 3.72 | | | 56.24 | 63.04 | | | | -2.47 |
| Pro7 | 4.68 | 2.31 1.89 | 2.02 | 3.79 3.64 | 61.51 | 30.66 | 27.26 | 50.50 | 3.40 | -1.83 |
| (4R)-FPro | 4.88 | 2.72 2.06 | 5.45 | 4.22 3.84 | 59.77 | 37.20 | 95.10 | 56.58 | -57.90 | -3.57 |
| Pro9 | 4.71 | 2.32 1.89 | 2.03 | 3.75 3.61 | 61.37 | 30.68 | 27.29 | 50.21 | 3.39 | -1.97 |
| Pro10 | 4.68 | 2.28 1.87 | 1.99 | 3.81 3.63 | 61.26 | 30.62 | 27.27 | 50.32 | 3.35 | -2.08 |
| Pro11 | 4.37 | 2.26 1.84 | 2.00 | 3.79 3.61 | 62.79 | 32.01 | 27.27 | 50.36 | 4.74 | -0.55 |
| Arg12 | 4.23 | 1.70 | 1.56 1.48 | 3.13 | 55.82 | 30.82 | 26.94 | 43.17 | | -0.96 |
| Val13 | 4.06 | 1.93 | 0.86 0.83 | | 61.82 | 33.04 | 20.41 20.98 | | | -0.72 |
| Tyr14 | 4.57 | 3.01 2.88 | | 7.10 Hε6.78 | 57.52 | 39.09 | | 53.21 Cε37.98 | | -0.66 |
| Lys15 | 4.27 | 1.84 | 1.34 | 1.63 Hε2.93 | 55.23 | 32.74 | 24.61 | 28.87 Cε41.82 | | -1.73 |

When comparing the two peptides, $^1$H and $^{13}$C chemical shifts of each type of FPro (4R or 4S) turn out nearly identical, inde-

pendent of their position in the sequence, suggesting that the local conformation of the pyrrolidine ring is not sensitive to the

sequence context, but is dictated by the fluorination stereochemistry at the Cγ-centre. To confirm this, we measured $^1$H-$^{19}$F

and $^1$Hα-$^1$Hβ scalar couplings within the FPro residues (Table 2). Heteronuclear $^1$H-$^{19}$F couplings were measured in 2D $^1$H-

$^1$H TOCSY spectra using the E.COSY cross-peak pattern $^1$Hγ-$^1$Hβ and $^1$Hγ-$^1$Hδ correlation peaks, while the homonuclear

$^1$Hα-$^1$Hβ couplings were measured using SERF experiments (see experimental section) (Fig. 3). These latter couplings are

diagnostic of the ring pucker, and a visual inspection of the coupling patterns observed for MpRS and MpSR peptides im-

mediately indicates that both (4R or 4S) FPro retain their *exo* or *endo* ring pucker in the context of a polyproline segment.

These scalar couplings are compared with the literature values determined for the free amino acid (Table 2) (Gerig et al.,

1973), and turn out to be very similar, except for the $^3$J$_{FH-3}$ coupling in (4S)-FPro where a difference of about 5 Hz is seen.

The reason for this is unclear, but could be due to the presence of either a neighbouring amide or amine group in the peptide

or free amino acid, respectively. Thus, it can be concluded that the strong bias of the five-membered ring conformation in-

troduced by 4-monofluorination (Gerig et al., 1973; DeRider et al., 2002) is fully preserved within the oligoproline context.

Using density functional theory (M06/cc-pVDZ in SMD water), we previously calculated for Ac-FPro-OMe that the Cγ-

*exo*:Cγ-*endo* population ratios are 93:7 for (4R)-FPro and 1:99 for (4S)-FPro (Hofman et al., 2018; Hofman et al., 2019). For

the purpose of NMR conformation and dynamics analysis, it is thus fair to assume that only one ring conformer is present. It

is known that proline normally interconverts between the Cγ-*exo* and Cγ-*endo* ring conformations within oligoprolines while

adopting a PP-II helix (Horng et al., 2006; Wilhelm et al., 2014). Similar to the concept of conformational frustration (Fer-

reiro et al., 2014), it can be stated that proline fluorination creates a form of "dynamic frustration" within the polyproline

helix.





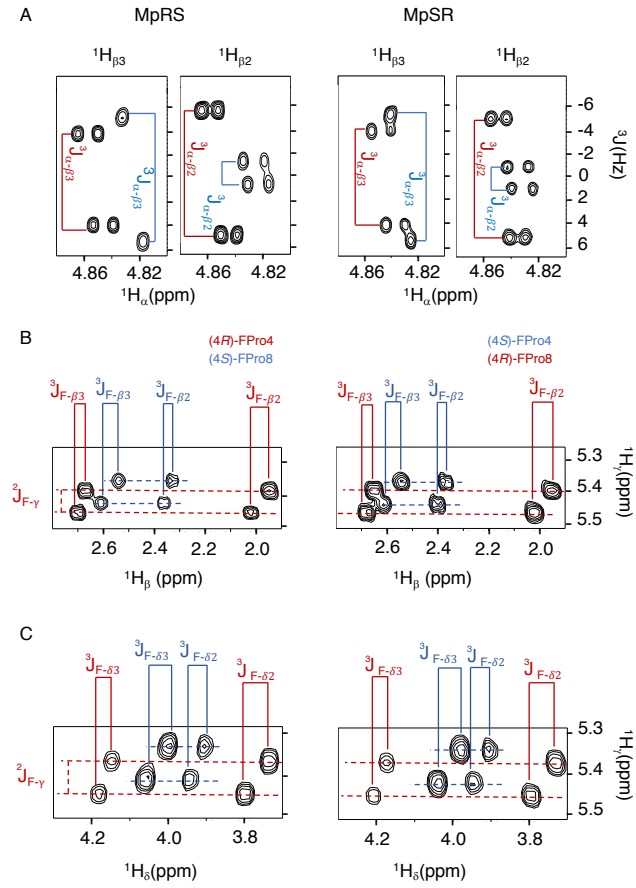


**Figure 3**: Puckering analysis of fluoroprolines. **A:** Homonuclear coupling constants $^3J_{H\alpha-H\beta2}$ and $^3J_{H\alpha-H\beta3}$ of the (4$R$) (in red) and (4$S$) (in blue) fluoroprolines in MpSR and MpRS peptides measured from SERF experiments at 298 K. **B:** The $^3J_{F-H\beta}$ heteronuclear coupling constants extracted from 2D $^1H$-$^1H$ TOCSY spectra using the E.COSY cross-peak pattern. **C:** The $^3J_{F-H\delta}$ heteronuclear coupling constants extracted from 2D $^1H$-$^1H$ TOCSY spectra using the E.COSY cross-peak pattern.




**Table 2**: Comparison of the scalar coupling constants (in Hz) of (4*R*)-FPro and (4*S*)-FPro measured in MpRS and MpSR peptides with those reported for the free fluoroproline residues (Gerig et al., 1973).

| | MpRS | | MpSR | | Free amino acid | |
|---|---|---|---|---|---|---|
| | P4 (4*R*) | P8 (4*S*) | P4 (4*S*) | P8 (4*R*) | (4*R*)-FPro | (4*S*)-FPro |
| $^3J_{F-\beta2}$ | 42.3 | 21.0 | 20.9 | 42.1 | 40.5 | 20.5 |
| $^3J_{F-\beta3}$ | 18.8 | 42.5 | 43.1 | 18.9 | 19.6 | 41.9 |
| $^3J_{F-\delta2}$ | 38.2 | 24.6 | 24.7 | 38.3 | 37.4 | 19.4 |
| $^3J_{F-\delta3}$ | 21.7 | 35.2 | 35.4 | 21.9 | 20.1 | 37.6 |
| $^3J_{\alpha-\beta2}$ | 10.1 | 3.0 | 3.1 | 10.2 | 10.4 | 2.8 |
| $^3J_{\alpha-\beta3}$ | 8.1 | 10.5 | 10.2 | 8.1 | 8.1 | 10.5 |


For the non-fluorinated prolines, the [13]C chemical shifts are mostly similar in the MpRS and MpSR peptides (Table 1), suggesting that the overall conformational properties of the peptide are not greatly affected by the permutation of the two FPro residues. It is also observed that the insertion of the PPII destabilizing (4*S*)-FPro within both polyproline segments does not alter the intensities of the strong Hα(*i*-1) to Hδ(*i*) NOE cross-peaks observed between all prolines of the segment including

the (4*S*)-FPro. In addition, the chemical shift differences between [13]Cβ and [13]Cγ are found within 5 ppm for all eight natural prolines, indicating a *trans* conformation of the Xaa-Pro peptide bond (Table 1) (Schubert et al., 2002). All of this indicates that the overall PPII secondary structure of a polyproline segment is maintained, regardless of the conformational bias of the individual fluoroproline residue. Nevertheless, subtle [13]C chemical shift differences between both peptides are observed in the prolines neighboring the FPro residues (3, 5, 7 and 9) (Table 1), with the most pronounced differences seen in the Cδ

chemical shifts (Fig. 3). Indeed, it has been shown for Ac-FPro-OMe model compounds that the (4*R*)- and especially the (4*S*)-FPro residues change the preferred $\psi$ dihedral angle (DeRider et al., 2002). The FPro residues thus appear to cause small local conformational equilibrium or dynamics changes in the local PP-II helix backbone, and further detailed conformational analysis is ongoing to confirm and quantify this effect. Finally, a minor set of peaks for prolines 10 and 3 is observed in the MpRS peptide in the 2D [1]H-[13]C HSQC-NOESY spectrum (Fig. 2); their origin could not be established.

The [19]F NMR spectra of each peptide are shown in Fig. 4. The assignment of the [19]F resonances can be made by comparing their chemical shifts with those of the Ac-FPro-Ome model compounds (*ca.* −178 ppm for (4*R*)-Fpro, −173 ppm for (4*S*)-Fpro) (Hofman et al., 2019). Just as for [1]H and [13]C chemical shifts, [19]F chemical shifts of each type of FPro change only slightly between peptides. Interestingly, several smaller peaks are found near the main peak of the (4*R*)-FPro at position 4 of the MpRS peptide, with one accounting for 35% of the total signal integral. Just as for the minor peaks in the [1]H-[13]C HSQC

(Fig. 2), these most likely correspond to minor forms of the peptide where a single proline or fluoroproline is in the *cis*-form. For oligoproline sequences, the *cis* form of internal prolines are known to be typically populated at a few percent, while the N- and C-terminal proline can have populations above 10% (Best et al., 2007; Urbanek et al., 2020). This illustrates the remarkable sensitivity of fluorine to its chemical environment, as it is able to resolve not just the *cis*-form of the FPro residue itself, but also of nearby proline residues within the oligoproline. However, it remains hard to identify which particular mi-

nor [19]F resonance belongs to which particular proline *cis*-form.



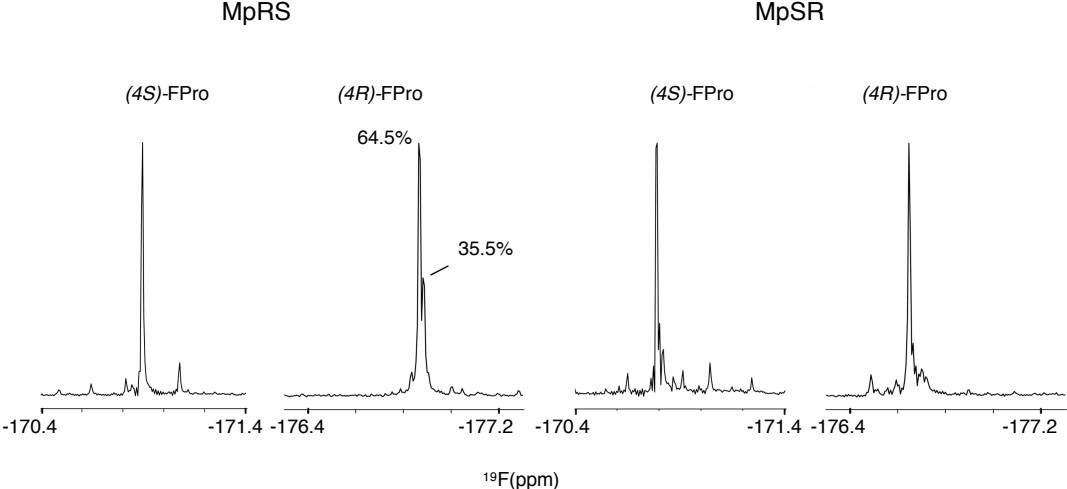

**Figure 4**: 1D $^{19}$F NMR spectra showing the fluoroprolines signals of the two model peptides MpRS on the left and MpSR on the right. The spectra were recorded at 298 K and 600 MHz. The (4*R*)-FPro within MpRS resonance displays a minor state accounting for 35.5% of the total peak integral.


### 2.2 $^{19}$F relaxation and dynamics

Spin relaxation rates are a useful source of information on molecular structure and dynamics. However, $^{19}$F relaxation theory is rather complex, with multiple dipole-dipole (DD) interactions to neighboring protons, strong chemical shift anisotropy (CSA), and a multitude of cross-correlations (Dalvit et al., 2017; Lu et al., 2019). This stands in contrast with protein back-
bone $^{15}$N relaxation where the dominant DD interaction with a single proton and $^{15}$N CSA is well understood. A quantitative analysis of $^{19}$F relaxation rates for both the (4*R*)– and (4*S*)–FPro residues in terms of dynamics thus requires knowledge of the various $^{1}$H-$^{19}$F distances within the fluoroproline structure, and also of the $^{19}$F CSA tensor. These were obtained (Table 3) using density functional theory for the energy minimum structures of Cγ-*exo* and Cγ-*endo* ring conformations in the *trans* form of N-acetyl-FPro–NMe$_2$, where the capping groups were chosen to emulate the oligoproline peptide context. In each
case, multiple protons had sufficiently small distances to the $^{19}$F nucleus in order to significantly contribute to DD relaxation. While the distance to the Hγ proton remains constant (2.0 Å), the distances to Hβ and Hδ protons change with conformation (Table 3) (Gerig et al., 1973). Proline ring conformation also has a profound effect on the anisotropy parameter Δσ of the chemical shift tensor: the major conformers (Cγ-*exo* for 4*R*-FPro and Cγ-*endo* for 4*S*-FPro) have Δσ ≈ –80 ppm, while the minor conformers have Δσ ≈ –30 ppm.

It has been shown that the proline ring pucker exchange occurs on a picosecond time scale (London et al., 1978; Sarkar et al.; 1986; Kang et al., 2007), and thus faster than the overall tumbling of a peptide of protein. Fluorine relaxation rates will be sensitive to this internal motion. However, because DD and CSA interactions vary in a correlated way with this motion, standard rotational diffusion autocorrelation functions cannot be used. Fortunately, as shown in the previous section, within the polyproline context both FPro residues adopt one dominant conformation. Although the work reported here can proceed,
it would be beneficial in the long run to design case-specific relaxation models that involve a picosecond scale correlated switch in the spin Hamiltonian parameters.



Theoretical calculations of all relaxation rates were performed using the brute-force numerical implementation of Bloch-Redfiedl-Wangsness relaxation theory available in *Spinach 2.6*, which automatically accounts for all dipole-dipole interactions, all chemical shift anisotropies, and all of their cross-correlations (Hogben et al., 2011). Molecular geometries and the relevant magnetic parameters (chemical shielding tensors and *J*-couplings) were imported from DFT calculations. The molecules in question are small enough that no spin system truncation is necessary. Longitudinal $^{19}$F relaxation rates were calculated for rotational correlation times between 10 ps and 100 ns at 14.1 Tesla (Fig. 5A), for both *exo* and *endo* conformers to evaluate the impact of proline ring pucker. The correlation time dependence of longitudinal relaxation rates for the major conformers of all FPro residues shows a peculiar 'camel hump' shaped profile, with two maxima at 0.3 ns and at 4.4 ns. The same picture was reported earlier for fluorinated aromatic amino acids based on a simplified relaxation model (Dalvit et al., 2017), it is caused by CSA and dipolar $^{1}$H-$^{19}$F relaxation contributions being maximal at different frequencies, namely the fluorine Larmor frequency ($\omega_F$) and the difference between proton and fluorine Larmor frequencies ($\omega_H - \omega_F$), respectively. For the minor conformers, the maximum at 0.3 ns is much lower than the one at 4.4 ns due to the lower CSA (Table 3). This demonstrates that, for longitudinal $^{19}$F relaxation, the contribution of motions operating at time scales up to about 3 ns are strongly influenced by the ring pucker distribution. When assuming only the major ring pucker to be present – as in the polyproline context – $R_1$ shows very little contrast in the 0.1 ns to 10 ns range, implying it is not an appropriate parameter to unambiguously probe dynamics. The experimental relaxation rates measured for MpRS and MpSR peptides at 298K are reported in Table 4; they fall into a narrow range between 2.1 and 2.3 s$^{-1}$, in agreement with the calculated values within the aforementioned correlation time range.

Transverse $^{19}$F relaxation rates (measured using the CPMG sequence) show the usual monotonic increase with the rotational correlation time (Fig. 5B). The difference in CSA between *exo* and *endo* puckers has a clear impact throughout, thereby complicating its interpretation in situations where puckers would be exchanging. To assess the contribution of slow motions, transverse relaxation rates were also measured using a spin echo (Table 4). This revealed about double values throughout, revealing exchange contributions at both sites for both MpRS and MpSR peptides. As residual exchange contributions cannot be excluded in the CPMG experiment, an interpretation of transverse relaxation rates would also be unreliable.

In contrast to $R_1$ and $R_2$, $^{1}$H-$^{19}$F cross-relaxation rates within the same carbon centre are purely dipolar and therefore likely to be easier to analyse. The $^{1}$Hγ-$^{19}$F NOE is ideal because Hγ has a distinct chemical shift at 5.6 ppm, allowing selective RF irradiation without perturbing the remaining protons of the proline ring; $^{1}$Hγ-$^{19}$F distance is independent on ring pucker. Fig. 5C shows the calculated steady-state $^{19}$F NOE upon $^{1}$Hγ saturation as a function of rotational correlation time. Just as for the $R_1$ curves, at long correlation times nearly identical curves for both the (4*R*)- and (4*S*)-FPro residues in each pucker are found, while at short correlation times a small difference is found between the puckers due to the dissimilar CSA. Importantly, the sigmoidal transition parts between fast-motion and slow-motion limits are similar in all four cases, making $^{1}$Hγ-$^{19}$F NOE a reliable parameter sensitive to motions with correlation times between 0.1 and 4 ns.

Experimentally, $^{19}$Fγ signal intensities were measured for several Hγ selective irradiation times, leading to the observation of NOE build-up curves that were fitted with a single exponential function to extract the cross-relaxation rates (Fig. 5D-E). For both peptides, the steady-state NOE ranges from –6.8 % at position 4 to –19.9 % at position 8 (Table 4), indicating faster dynamics experienced by the first polyproline segment compared to the second. These values correspond to rotational correlation time estimates of 0.5 ns for the proline at position 4, and 0.8 ns for the proline at position 8.



**Table 3**: Internuclear distances between $^{19}$F atom and the neighboring protons for representative conformers of major and minor conformations of the *(4R)-* and *(4S)-* FPro and corresponding $^{19}$F CSA tensor parameters derived from Gaussian calculations. Δδ is the chemical shift tensor anisotropy, η is the asymmetry parameter, and

265 $\delta_{xy}^{anti}, \delta_{xz}^{anti}, \delta_{yz}^{anti}$ are the antisymmetric components to the full CSA tensor in the principal axes coordinate system of the symmetric part of the tensor.

| | Distances (Å) | | | | | $^{19}$F CSA tensor | | | | |
| | F-Hγ | F-Hβ2 | F-Hβ3 | F-Hδ2 | F-Hδ3 | Δδ (ppm) | η | $\delta_{xy}^{anti}$ (ppm) | $\delta_{xz}^{anti}$ (ppm) | $\delta_{yz}^{anti}$ (ppm) |
|---|---|---|---|---|---|---|---|---|---|---|
| *(4R)*-exo major | 2.03 | 3.29 | 2.56 | 3.3 | 2.5 | -74.2 | 0.120 | 4.71 | 2.21 | -3.45 |
| *(4R)-endo* minor | 2.02 | 2.89 | 2.50 | 2.97 | 2.44 | -25.6 | 0.396 | 7.29 | 2.34 | 4.42 |
| *(4S)-endo* major | 2.01 | 2.49 | 3.29 | 2.4 | 3.25 | -84.9 | 0.392 | -3.26 | -4.27 | -6.01 |
| *(4S)-exo* minor | 2.02 | 2.49 | 2.89 | 2.52 | 2.88 | -33.3 | 0.483 | 5.78 | 2.20 | -2.32 |

**Table 4**: Experimental longitudinal and transverse relaxation rates together with the nuclear Overhauser effect measured for both peptides at 298 K on a 600 MHz spectrometer.

| | MpRS | | MpSR | |
| | P4(4R) | P8(4S) | P4(4S) | P8(4R) |
|---|---|---|---|---|
| R₁(s⁻¹) | 2.23± 0.04 | 2.2± 0.01 | 2.30± 0.01 | 2.13± 0.01 |
| NOE max (%) | -6.8 | -19.9 | -9.3 | -19.0 |
| ρ (s⁻¹) | 1.85 | 1.76 | 2.25 | 1.59 |
| σ (s⁻¹) | -0.12 | -0.33 | -0.2 | -0.28 |
| R₂(s⁻¹) Spin Echo | 20.3 ± 0.5 | 24.96 ± 0.6 | 12.4 ± 0.5 | 18.5 ± 0.4 |
| R₂(s⁻¹) CPMG | 8.6 ± 0.5 | 8.2 ± 0.3 | 5.4 ± 0.3 | 9.7 ± 0.5 |



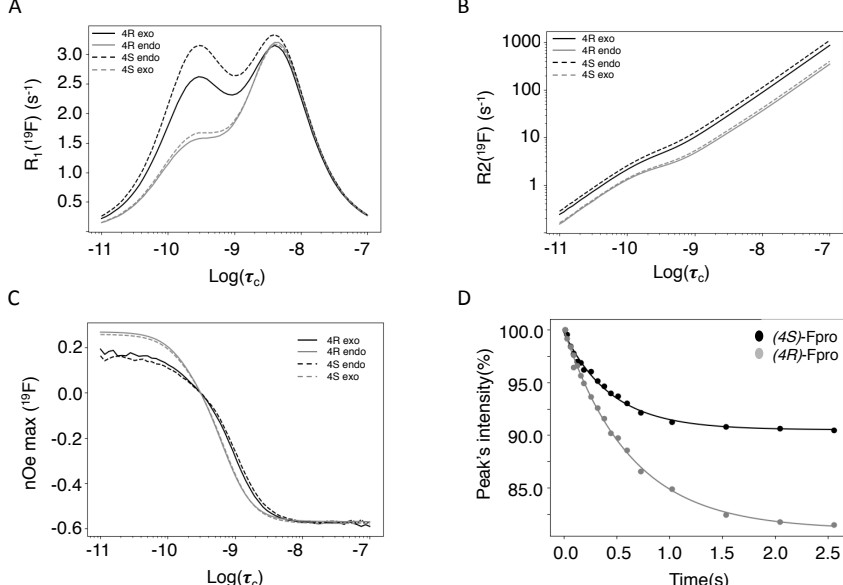

**Figure 5**: **A:** Calculated [19]F longitudinal relaxation rates as functions of rotational correlation time. **B:** Calculated [19]F transverse relaxation rates as functions of rotational correlation time. **C:** Calculated steady state fluorine-proton heteronuclear NOEs. Relaxation data were calculated for (4R)-FPro in the major Cγ-*exo* (black plain line) and minor Cγ-*endo* (grey plain line) and for (4S)-FPro for the major Cγ-*endo* (black dashed line) and the minor Cγ-*exo* (grey dashed line) conformations. **D:** Experimental NOE build-up at Fγ upon selective saturation of Hγ proton measured for the MpSR peptide at 298 K. The dots are the experimental peak intensities and the solid line is the corresponding fit to a monoexponential function.

### 2.3 Impact of proline modifications on the binding of SH3

SH3 domains are small modular protein domains of 50-70 amino acids that typically interact with proline rich motifs (PRM) and that are highly represented in the human genome (Saksela et al., 2012). Many experimental and theoretical studies have been conducted to decipher the molecular mechanisms underlying both binding affinity (in the 0.1-100 μM range of dissociation equilibrium constant $K_d$) and specificity of SH3 domains that primarily recognize PXXP sequence motifs. This mechanism involves the aromatic indole ring of the tryptophan 37 residue exposed at the surface of the SH3 domain that mediates CH•••π interaction with proline residues. Additional binding energy is provided by electrostatic interactions between the SH3 surface residues and those flanking the PXXP motif of the binding partner.

In order to measure the binding affinities between the Vinexinβ SH3.3 domain and the model peptides, a titration experiment was performed where increasing amounts of peptide were added to a solution of [15]N labeled SH3 domain. Apart from MpRS and MpSR peptides, a titration was also performed with a non-fluorinated reference peptide. Just as for most SH3-PRM interaction studies, a gradual frequency shift of a subset of [1]H-[15]N correlation peaks in the [1]H-[15]N HSQC spectra was observed, indicative of fast exchange between bound and free states of the protein (Fig. 6A). Under this exchange regime, the chemical shifts provide an accurate measure of the bound protein fraction, enabling the determination of an equilibrium



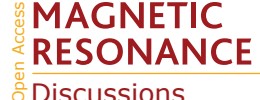

dissociation constant $K_d$ (*vide infra*). Interestingly, a striking difference between the peptides is observed in the $^1$H-$^{15}$N HSQC during the titration, where the trajectory of the tryptophan 37 $^{15}$N$_\varepsilon$ -$^1$H$_\varepsilon$ correlation appears different for the MpRS peptide compared to the MpSR peptide (Fig. 6B). Whether this reflects a direct interaction between the Trp37 aromatic ring with (4*S*)-FPro8 in the PXXP binding motif, or an alteration of the binding complex indirectly caused by the (4*S*)-FPro8

residue in MpRS will require further investigation.

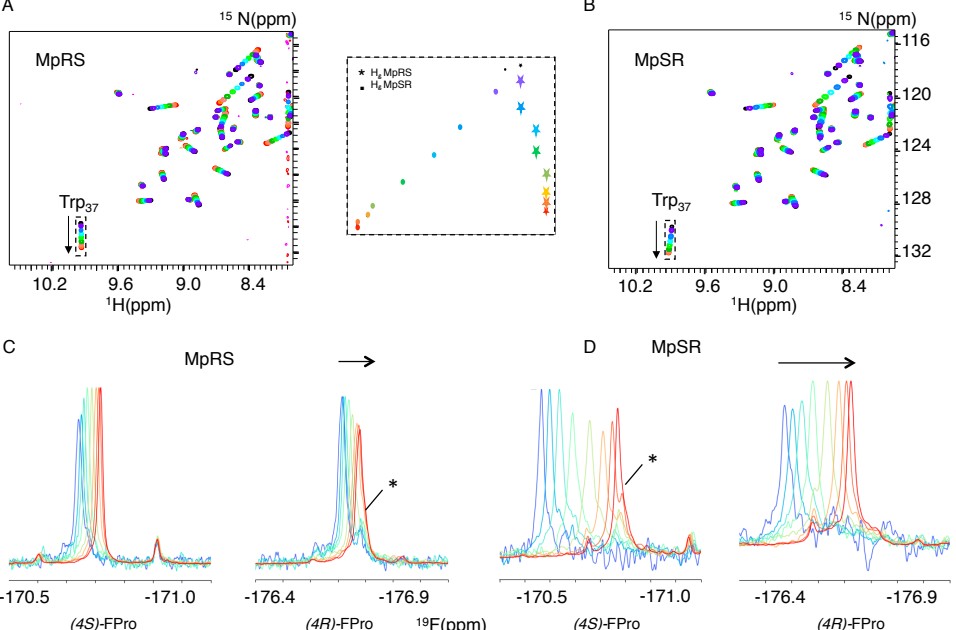

**Figure 6**: **A** and **B:** A series of $^1$H-$^{15}$N HSQC spectra of Vinexin β SH3.3 domain recorded upon successive addition of MpRS (A) and MpSR (B) peptides. The black spectrum corresponds to the first titration point where no peptide is added

while the red spectrum corresponds to the last titration point. The arrows indicate the tryptophan 37 N$_\varepsilon$-H$_\varepsilon$ cross-peak trajectories during the titration. The insert displays a zoom on these cross-peak trajectories shown as the position of the peak center for both MpRS (stars) and MpRS (disks) peptides. **C** and **D:** Series of 1D $^{19}$F spectra recorded during the same titration experiment of Vinexin β SH3.3 domain by MpRS (C) and MpSR (D) peptides. For the (4*R*)-FPro4 in MpRS and (4*S*)-FPro4 in MpSR, a minor peak that overlaps with the main peak at high peptide concentrations is indicated by an asterisk. At low

peptide concentrations (blue) the spectra are indicative of a mostly bound form while at high peptide concentrations (red), the spectra converge to those observed for the free peptides.

For the MpRS and MpSR peptides, the peptide-protein titration can also be observed using $^{19}$F NMR, allowing to simultaneously monitor the binding event from the perspective of the protein (receptor) and the peptide (the ligand) (Fig. 6C). Thanks

to the availability of a cryogenic fluorine probehead, the $^{19}$F signals could be detected even at the first titration point where the peptide concentration was just 50 μM and strong exchange broadening was present. Just like for the $^1$H-$^{15}$N chemical shifts, increasing the peptide concentration resulted in a gradual shift of the $^{19}$F resonances, indicative of a fast exchange regime. Interestingly, for both the signals from (4*R*)-FPro4 in MpRS and (4*S*)-FPro4 in MpSR, a minor peak is observed that





does not shift during the titration (highlighted by a star in Fig. 6C and 6D). This minor peak thus appears to belong to a state

that is not competent for SH3 binding. This proline is located in the first polyproline segment and this observation implies that at least two states of the complex are evidenced by the fluorine resonance at this position. At higher peptide concentrations, the peaks sharpen up with addition of peptide, which can be explained by the increasing fraction of the unbound peptide and thus lower amount of exchange broadening and faster tumbling correlation time. Visual inspection of the $^{19}$F spectra reveals that the extent of chemical shift perturbation for both $^{19}$F signals in each peptide appears similar, even though P8 falls

within the binding motif. When comparing the MpRS and MpSR peptides, it can be seen that the extent of chemical shift perturbations is the highest for the MpSR peptide, qualitatively already indicating the higher affinity of MpRS relative to MpRS.

Both the $^{1}$H/$^{15}$N chemical shift perturbations of the SH3 domain as well as the $^{19}$F chemical shift pertubations of the peptides can be used to assess the binding affinity. For this, the stoichiometry of the binding was first evaluated. Indeed, even though

a single canonical PXXPX+ motif is present in the peptide sequence imposing binding specificity, a closer inspection shows that multiple non-specific PXXP motifs can be identified (Fig. 1), potentially leading to additional ways for the SH3 domain binding. For this, two binding models were used where the peptide and SH3 domain can bind either up to a 1:2 ratio or only in a 1:1 ratio. Both the $^{19}$F and $^{1}$H/$^{15}$N chemical shift data were fitted simultaneously using these models. Based on the goodness of fit reported as the reduced $\chi^2$, the ternary complex turned out to be unnecessary to explain the data, thus implying

that only one SH3 binds to the peptide. The dissociation constants ($K_d$) found in this way were 96 ± 30 µM for MpSR and 273 ± 30 µM for MpRS. The reported uncertainties account for the uncertainty on protein and peptides concentration measurements that was estimated to be 15%. However, a strikingly good agreement was observed between the experimental and back-calculated $^{19}$F chemical shifts, with a standard deviation of only 0.6 Hz despite the large peak widths of 10-20Hz (Fig. 7A). This excellent precision thanks to the sparsity of the 1D spectrum highlights one important feature of $^{19}$F NMR spec-

troscopy to study molecular interactions. The fitted differences in bound and unbound $^{19}$F frequencies is about twice as high for MpSR (265 ± 8 Hz for (4$S$)-FPro4 and 218 ± 8 Hz for (4$R$)-FPro8) than for MpRS (88 ± 17 Hz for (4$S$)-FPro4 and 100 ± 17 Hz for (4$R$)-FPro8).

In addition, the $K_d$ value was also determined for the non-fluorinated peptide using the $^{1}$H/$^{15}$N chemical shift perturbations alone using a binding model with 1:1 stoichiometry, which was found to be 74 ± 25 µM, which is similar to the MpSR pep-

tide. It thus appears that the presence of (4$R$)-FPro within the binding motif has a negligeable effect on the interaction with SH3, while (4$S$)-FPro significantly lowers the binding affinity despite our observations that suggest a preserved PPII conformation.

The exchange line broadening during the titration experiment also reports on the binding kinetic. Thus, the major $^{19}$F peaks were fitted using a Lorentzian line shape and the line widths obtained in this way provide an estimate of the apparent trans-

verse relaxation constant $R_2^{\ddagger}$ as a function of peptide concentration (Fig. 7B and C), that can be used to derive the binding kinetics. A simplified expression of the exchange contribution to $R_2^*$ as a function of peptide and SH3 concentration was used that is valid for the fast exchange approximation ($k_{exc} \gg \Delta\omega$) (Kovrigin et al., 2012):

$$R_2^{\ddagger} = p_f R_2^f + p_b R_2^b + p_f p_b \frac{\Delta\omega^2}{k_{exc}} \tag{1}$$

with:

$$k_{exc} = k_{on}\left([SH3]_{free} + K_d\right) \tag{2}$$

where $p_b$ and $p_f$ are the bound and free fractions of the peptide, $R_2^b$ and $R_2^f$ are the transverse relaxation rates of the bound and the free forms, and $\Delta\omega$ the frequency difference between the bound and free states multiplied by $2\pi$. Taking the values





of $\Delta\omega$, $p_b$, $p_f$, $[SH3]_{free}$ and $K_d$ from the chemical shift perturbation fitting, the $R_2^b$, $R_2^f$ and $k_{on}$ values were subsequently fitted to the experimental $R_2^{\ddagger}$ values. For the MpSR peptide, the optimization was performed independently for the (4S)- and (4R)-FPro $^{19}$F signals, leading to a fairly good agreement between experimental and modeled values (Fig. 7B). This provided fitted association kinetic constants $k_{on}$ of $0.9\ 10^8 \pm 0.2\ 10^8\,\mathrm{M^{-1}s^{-1}}$ and $1.2\ 10^8 \pm 0.2\ 10^8\,\mathrm{M^{-1}s^{-1}}$ for the (4S)-FPro4 and (4R)-FPro8 signals, respectively. These values are consistent with a simple one-to-one association mechanism driven by a free

diffusion process of the two binding partners.

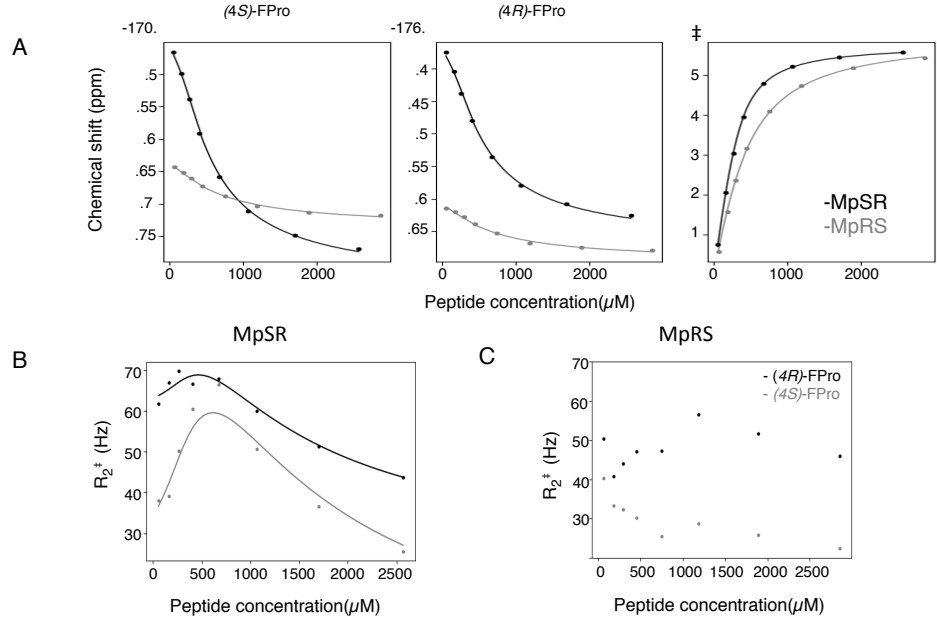

**Figure 7**: **A:** $^{19}$F chemical shift variation of (4S)-FPro (left) and (4R)-FPro (middle) in MpSR (in black) and MpRS (in gray) peptides extracted from 1D $^{19}$F spectra. Left panel displays the $^1$H, $^{15}$N composite chemical shift from $^1$H-$^{15}$N HSQC. The experimental data (dots) were fitted simultaneously to derive the equilibrium dissociation constants for the two pep-

tides.(solid lines). **B:** Variation of the apparent $^{19}$F transverse relaxation rates ($R_2^{\star}$) derived from the $^{19}$F line widths as a function of MpSR peptide concentrations. The grey and black lines indicate the expected variation if a fast on-rate kinetic is considered. **C:** Variation of the apparent $^{19}$F transverse relaxation rates ($R_2^{\ddagger}$) as a function of MpRS peptide concentrations.

For the MpRS peptide, the profile of $R_2^{\ddagger}$ as a function of peptide concentration showed a markedly different behavior. After

an initial sharpening of about 10 Hz for both $^{19}$F signals upon addition of the second peptide aliquot to the SH3 sample, a line broadening was observed for (4R)-FPro at position 4 while a continuous sharpening is experienced by the fluorine resonance of (4S)-FPro at position 8. This observation is peculiar, as in a simple one-binding site model one would expect a similar profile for both signals. This suggests a more complex binding mechanism involving at least one supplementary minor state. This is consistent with the observed significant reduction of the chemical shift differences between the bound

and free forms of the MpRS peptide compared to MpSR as noted previously. For the MpRS peptide, the combined analysis of fluorine and proton spectral properties is insufficient to specify a specific binding model. However, together with the



slight difference observed for the trajectory of the tryptophane $^{15}N_\epsilon$ -$^1H_\epsilon$ correlations in the $^1$H-$^{15}$N HSQC (Fig. 6A), this indicates that the structure or dynamics of the complex are altered by the insertion of (4*S*)-FPro within the canonical SH3 binding motif.


### 3. Discussion

(4*R*)- and (4*S*)-fluorinated prolines have, so far, been used in structural biology studies. This work demonstrates their hither-to neglected potential in biomolecular $^{19}$F NMR investigations. In contrast to fluorination of most amino acids used in such studies, proline fluorination changes its conformational and dynamic properties, leading to modified protein-protein interac-

tions. Although this may seem undesirable at first, this can be put to good use – as shown above – to modulate the interaction between a PRM and an SH3 domain. Using a model peptide containing two oligoproline sequences, permutations of two types FPro residues in conjunction with $^{19}$F NMR analysis allowed studying the consequences of the conformational biases on the binding equilibrium with the SH3 domain. While the binding affinity appears unaltered by the introduction of (4*R*)-FPro at position 8 that lies within the SH3 binding motif, the insertion of (4*S*)-FPro at the same position leads to a substantial

decrease of the binding affinity. Similar conclusions were drawn in studies involving SH3 domains of cortactin and human hematopoietic-lineage cell-specific protein 1, where insertions of (4*R*)- or (4*S*)-FPro residues in the cognate PRM weakened the binding affinity (Ruzza et al., 2006; Borgogno et al., 2013). Interestingly, these and other (Horng et al., 2006) studies used circular dichroism spectroscopy to confirm that PPII conformational preference is stabilized by (4*R*)-FPro, meaning that the expected associated increase in SH3 binding affinity is negated by other effects introduced by the presence of the

fluorine. Ruzza *et al* (Ruzza et al., 2006; Borgogno et al., 2013) suggested this could be due to a destabilization of the hy-drogen bond formed by the proline's carbonyl group due to the inductive effect of fluorine, or by destabilization of proline's interaction with aromatic side-chains of the SH3. Further studies are required to disentangle these effects.

Apart from binding affinities, $^{19}$F line shape analysis allowed kinetic information to be extracted. Thanks to the exquisite susceptibility of the $^{19}$F signal line width to chemical exchange phenomena, it was found that the binding on-rate of the SH3

domain is fast and diffusion-limited. This result is consistent with a recent study reporting diffusion limited binding kinetics of an SH3 domain to a PRM peptide, where a fluorinated tryptophan inserted in the SH3 domain allowed simultaneous mon-itoring of $^{19}$F and $^1$H/$^{15}$N chemical shift perturbations measured from the SH3 domain (Stadmiller et al., 2020). The differ-ence with our study is that the $^{19}$F chemical shift perturbations report on the binding event from the point of view of the bind-ing peptide, providing complementary information with the $^1$H/$^{15}$N chemical shift perturbation from the SH3 domain. Here,

observation of fluorine resonances perturbations on the ligand evidenced a different dynamics of the SH3 domain on the polyproline peptide upon introduction of the conformationally biaised (4*S*)-FPro in the cognate PRM.

Although the available numerical tools allow in principle to model spin relaxation processes in multi-spin systems very accu-rately, a major complication is the picosecond scale dynamics of the five-membered ring, mainly due to the strong depend-ence of $^{19}$F CSA on ring pucker. This effect can be mitigated by a strong ring pucker bias, as is typically the case for (4*R*)-

and (4*S*)-FPro residues. However, in general, and especially for other FPro variants without pucker bias (Hofman et al., 2018), a more advanced theoretical analysis will be required. Still, the measurement of the NOE between the geminal Hγ and Fγ provided an interesting way to probe local dynamics with correlation times between 0.3 and 4-5 ns.

The comparison of transverse relaxation at two different effective B$_1$ fields revealed the presence of motions occurring at the μs to ms time-scale. It should be noted that the presence of many $^1$H-$^{19}$F couplings within the FPro spin system implies that

recently developed $^{19}$F relaxation dispersion experiments cannot be applied (Overbeck et al., 2020). The dispersion of proton frequencies in the fluorinated prolines enable their selective excitation, a feature that was exploited for the selective Hγ-Fγ NOE and can be further used to develop sequences adapted to FPro spin systems. The molecular origin of the difference in



dynamics between the oligoproline segments remains unclear, and suggests that the flanking amino acid sequences can play a role in the conformational and dynamical preferences of polyproline segments. Importantly, given the absence of amide

protons and the low $^1$Hα and $^{13}$Cα chemical shift dispersion, this information would be very difficult to obtain from $^1$H, $^{13}$C or $^{15}$N measurements.

In conclusion, fluorinated prolines provide an attractive tool for biomolecular NMR studies, in addition to their well-established application of controlling proline conformation. Given the increasing capabilities of chemical biology techniques that allow introduction of unnatural amino acids in proteins, such as chemical ligation or genetic code expansion

(Debelouchina et al., 2017), we foresee that $^{19}$F NMR studies through FPro residues will find their way to larger protein constructs. Apart from (4*R*)- and (4*S*)-fluorinated prolines, many more mono- and difluorinated prolines have been described (Verhoork et al., 2018), providing a rich set of fluorine labelling options for PRMs that can be tuned to the specific needs in terms of conformational control and/or $^{19}$F NMR properties. Further investigations in this respect are ongoing. There exists a broad scope of SH3 and other domains that interact with PRMs, where the weak binding affinities are determined by detailed

structural features of the binding motifs and altered by post-translational modifications or physiological conditions, leading to finely regulated protein-protein interaction network. We are confident that fluorinated prolines and $^{19}$F NMR provide an elegant way to address the complexity of these systems.




## 5. Material and Methods

*Sample preparation*

The MpRS and MpSR peptides were produced by solid phase synthesis using Fmoc-amino-acids using a model 431A pep-
tide synthesizer from Applied Biosystems (Foster City, CA, USA). Fmoc-protected (4*R*)- and (4*S*)-FPro amino acids were
purchased from Bachem SA. Peptides were purified by reversed-phase HPLC, and checked by electrospray ionization-time
of flight mass spectrometry (ESI-TOF). The Vinexin β SH3.3 was obtained using recombinant expression of a GST fusion
protein in *E. coli* using pGEX plasmids as described (Lalevee et al., 2010). After thrombin cleavage, the protein was purified
using size exclusion chromatography and eluted with phosphate buffer (40 mM Phosphate, NaCl 100 mM, DTT 2 mM, pH
7). Before titration experiments, a dialysis was performed using 1 kDa  and 3 kDa cutoff membrane for the peptide and the
protein, respectively, and a common dialysis bath containing the buffer used in interactions experiments. Protein concentra-
tions were determined by measuring the OD at 280 nm (molar absorption coefficient 11460 $M^{-1}.cm^{-1}$). Peptide concentra-
tions were measured by $^1$H NMR by comparing the integrals of peptide resonances with those of tryptophan of known con-
centration in a sample containing small amounts (10-30 μM) of both compounds in $D_2O$ as described (Kohler et al., 2015).
For assignments, lyophilized powder of MpRS and MpSR peptides were dissolved in 170 μL of $D_2O$ for a final concentra-
tion of 1 mM in 3 mm tubes.

*NMR experiments*

$^{19}$F NMR spectra were recorded on a Bruker Avance I spectrometer operating at a $^1$H frequency of 600 MHz and equipped
with a cryogenic QCI-F probe. $^1$H and $^{13}$C spectra were recorded using a Bruker Avance III spectrometer operating at a $^1$H
frequency of 700 MHz and equipped with a cryogenic TCI probe. Standard full-range $^1$H-$^{13}$C HSQC (10 ppm $^1$H x 80 ppm
$^{13}$C) were recorded on MpRS and MpSR peptides for the carbon assignment. The number of points in the time domain was
4096 in $F_2$ and 4096 in $F_1$.  In addition a high resolution 2D $^1$H-$^{13}$C HSQC-NOESY was recorded with the same $^1$H spectral
width, but with a narrow carbon bandwidth of 3 ppm, centered on the proline's Cδ resonances (47.3 ppm). The number of
points in the time domain was 1024 in $F_2$ and 256 in $F_1$.  The resulting resolution in the $^{13}$C dimension was 4 Hz/pt. During
carbon evolution, a gradient-enhanced frequency-selective $^{13}$C 180° refocusing pulse (4 ms RSNOB) was applied to avoid
interference from folded peaks from outside the spectral region. The NOESY mixing time was 80 ms. The relaxation time
was set to 1 s and 300 transients were recorded for each $t_1$ point resulting into a total experiment time of 1 day and 4 hours.
The $^1$H-$^{19}$F heteronuclear coupling constants were measured from the $^1$H-$^1$H TOCSY spectra (MLEV spinlock 80 ms) rec-
orded at 700 MHz.  The spectral width was 10 ppm in both $F_1$ and $F_2$, with the number of time domain points 4096 in $F_2$ and
512 in $F_1$, resulting in resolutions in $F_1$ and $F_2$ of 23.4 and 2.9 Hz/pt, respectively. The $^1$H-$^1$H couplings were measured using
SERF experiments (Fäcke et al., 1995) modified to use the Pell-Keeler method (Pell et al., 2007) to obtain 2D absorption
mode line shapes, as recently proposed (Sinnaeve, 2007). The active spin refocussing selective 180° pulse was a RE-BURP
pulse of 14.5 ms set to invert just the FPro Hα signals, while the selective inversion 180° pulses were I-BURP pulses of
12.85 ms set to invert just one Hβ proton per FPro residue at a time. The spectral width was set to 1.07 ppm in $F_2$ and 23.5
Hz in $F_1$, with the number of time domain points 1024 in $F_2$ and 64 in $F_1$, resulting in resolutions in $F_1$ and $F_2$ of 0.7 and 1.5
Hz/pt, respectively.

$^{19}$F $R_1$ and $R_2$ relaxation parameters were measured at 600 MHz ($^1$H frequency) and 298 K using standard inversion recovery
and CPMG experiments respectively. The carrier frequency was set to -174 ppm with a spectral width of 12 ppm and an
interscan relaxation delay of 4 s. The inversion recovery relaxation build-up delays ranged from 1 ms to 3 s with an expo-
nential sampling with one point repeated for uncertainty estimation, resuling in 20 data points in total. The CPMG sequence



was measured using a half-echo delay of 200 μs, or as a single spin echo with variable delays. A 180° proton pulse was applied every 2.8 ms at the fluorine echo time to average cross-correlation effects and ensure a single exponential decay (Farrow et al., 1995). Sampled relaxation delays ranged from 1 ms to 460 ms, with 16 data points in total.

[1]H-[19]F nOe buildup experiments were measured by selectively saturating the Hγ proton, using a train of sinc shaped soft 180° pulses centered at 5.42 ppm. The pulse duration was 2.8 ms and was applied every 4 ms prior to fluorine acquisition. The saturation times ranged from 10 ms to 2.6 s, including one repeat for error estimation, resulting in 16 data points in total. Processing of 1D [19]F spectra and quantification were performed using an open-source Python package dedicated to Fourier spectroscopies called "Spectrometry Processing Innovative Kernel" (SPIKE) (Chiron et al., 2016). An exponential line broadening of 8 Hz was applied prior to Fourier Transform for signal apodization. Line fitting was done using the least-square minimizer of the Scipy optimize toolbox to find the optimal set of the signal parameters minimizing the squared differences between the experimental and calculated spectra. 2D spectra used for peptide assignments were processed using Topspin 2.6 (Bruker) and visualized in CcpNmr Analysis V2 (Vranken et al., 2005). Relaxation parameters were obtained by fitting relaxation data to a three parameters single exponential model using the least-square algorithm implemented in the Scipy optimize toolbox (Levenberg-Marquart).

The selective longitudinal relaxation rate constants $\rho$ and the proton-fluorine cross-relaxation rate $\sigma$ were obtained by identification of the three optimized parameters to the following equation:

$$I(t) = I_0 + \frac{\sigma}{\rho}\frac{\gamma_H}{\gamma_F}I_0(1 - e^{-\rho t}) \qquad (3)$$

where $I_0$ is the equilibrium signal intensity and $\gamma_H$, $\gamma_F$ are the proton and fluorine magnetogyric ratios, respectively.

*Electronic structure theory and spin relaxation theory*

All electronic structure theory calculations were performed using *Gaussian09* (Frisch et al., 2009). Molecular geometries of proline isomers and conformers were optimised for fluoroproline moieties (capped with an acetyl group on the NH side and a dimethylamino group on the COOH side) using density functional theory with the M06 exchange-correlation functional (Zhao et al., 2008) and cc-pVDZ basis set (Peterson et al., 2002) in SMD chloroform (Marenich et al., 2009). Hessians were checked for positive definiteness at convergence point, and magnetic property calculations (shielding tensors and *J*-couplings) then proceeded using gauge-independent atomic orbital method (London et al., 1937) with the basis set decontracted and augmented with tight Gaussian functions (Deng et al., 2006) for the calculation of isotropic *J*-couplings.

Spin relaxation theory calculations were performed using *Spinach 2.6* (Hogben et al., 2011). Cartesian coordinates, chemical shielding tensors, and *J*-couplings of all fluorine and hydrogen atoms were imported from *Gaussian09* logs, and a numerical evaluation (Goodwin et al., 2015) of Redfield's relaxation superoperator (Redfield et al., 1957) for the resulting 16-spin system was carried out using the restricted state space approximation (Kuprov et al., 2007; Edwards et al., 2014), (IK-1(4,4) basis set) with a 5 Angstrom distance cut-off for dipolar interactions. Rigid-molecule isotropic rotational diffusion approximation was used. Longitudinal and transverse relaxation rates for the spins of interest were extracted as the matrix elements of the relaxation superoperator corresponding to Lz and L+ states of those spins. The implementation of Bloch-Redfield-Wangsness theory in *Spinach* automatically accounts for all applicable cross-relaxation and cross-correlation effects (Kuprov et al., 2011).

*Titration experiments*



Titrations were performed by successive addition of stock peptide solutions to a sample of SH3 protein at 314 µM in a 3 mm NMR tube. In order to reduce the dilution of the initial protein solution and keep the aliquot volumes within values compatible with low pipetting errors (1 to 3 µL), initial aliquots were added using stock solutions diluted by a factor 2. The concentrations of the stock solutions were 5.1 mM and 5.7 mM for MpSR and MpRS, respectively. For every titration point, a 1D
$^{19}$F spectrum and a $^1$H-$^{15}$N-HSQC were recorded at 298K on the same spectrometer, taking advantage of the QCI-F probe. The protein chemical shift perturbation was averaged over 9 $^1$H-$^{15}$N correlations that displayed a similar apparent titration profile (from the amino-acids Q14, N15 (side chain N$_{\delta2}$), D17, L21, W37 (side chain N$_\varepsilon$), V39, G49, T50, V56).

The composite chemical shift was calculated using:


$$\Delta\delta = \sqrt{(\Delta\delta_N)^2 + (5\Delta\delta_H)^2}$$   (4)

where $\Delta\delta_N$ and $\Delta\delta_H$ are the $^{15}$N and proton chemical shift difference measured between the free protein and the protein in presence of a given amount of peptide.

The modeling of the interaction was performed using an in-house Python script that solves the equilibrium concentrations of a set of interacting molecules by integrating the set of coupled differential equations until steady state is reached (https://github.com/delsuc/SpinEq). To fit the experimental data, we used the fluorine frequencies of free and bound states, and the frequency of the bounded SH3 as parameters. Depending on the model, one or more equilibrium constants were
given. The goodness of fit was assessed using the reduced chi2:

$$\kappa^2 = \frac{1}{N-NP}\sum_{i=1}^{N}\left(\delta_i^{exp} - \delta_i^{calc}\right)^2$$   (5)

**Aknowledgements**
This work was supported by the Agence Nationale de la Recherche (FLUOVIAL ANR-18-CE44-0009-01, FLUOPROLINE ANR-20-CE11-0025) the French Infrastructure for Integrated Structural Biology (FRISBI, ANR-10-INSB-05-01), Instruct-ERIC, the Centre National de la Recherche Scientifique, University of Strasbourg. The Research Foundation – Flanders (FWO) is indebted for a research project to J.C.M. and D.S. (3G011015), a PhD fellowship to E.O. and staff exchange fund-
ing (FWO-WOG Multimar). The EPSRC is thanked for a partial PhD grant to G.-J.H. (EPSRC-DTG EP/M50662X/1) and instrument funding (core capability EP/K039466/1). A.B.B is supported by a fellowship from the région Grand Est and ANR. Claude Ling is warmly acknowledged for maintenance of the NMR spectrometers.




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
