# Peer review of "Fluorine NMR study of proline-rich sequences using fluoroprolines"

_Magnetic Resonance, 2021_

## Referee Comment (RC2)

The authors present a detailed NMR investigation of two SH3 class II binding peptides, containing each a (4R)- and (4S)-fluoroproline but at different location. They first present the NMR assignment of the peptides, and exploit a high resolution NOESY-HSQC spectrum to assign all proline resonances.

In this aspect, I am suprised why they only give a single value for the $H_\delta$ protons, knowing that the two $H_\delta$'s can be distinguished, and actually give infomation on the flexibility of the residues (see for example Ahuja et al. JMB 2016). I also wonder whether the larger dispersion of the Pro $C_\delta$ carbons in the MpRS peptide compared to that of the MpSR peptide (Figure 2) has a meaning ? When I compare the $C_\delta$ spread of the two prolines flanking the 4R-FP in the MpRS peptide ($\Delta C_\delta$(3-5) = -0.3ppm) with the same value in the MpSR peptide ($\Delta C_\delta$(7-9) = +0.27ppm), I again wonder whether the chemical shift contains structural information.

Not only the ring pucker but also the backbone conformation of the proline is influenced by the fluorine incorporation, with the (4S)-FP favoring the cis conformation. Here, I am somewhat confused. If the fluorine spectrum of the (4R)-FP4 in MpRS shows a major and minor peak in a 1 :3 ratio, what do they represent ? A major trans form and a minor cis form of this floroproline? But these should then also show up in the 1H-13C HSQC spectra, and the cis form should be characterized by a $H_\alpha$-$H_\alpha$ cross peak ? And is the situation different for the (4R)-FP8 in the MpSR peptide ? What about the (4S)-FP in both peptides ? Are they in the cis conformation ? Elucidating these points seems important for the interested reader.

In order to characterize the movements of the FP rings, they turn to relaxation measurements. These are not easy to interpret, knowing the multiple dipolar terms and the important csa contribution. With the help of Spinach simulations, they obtain reasonable estimates for the different rates as a function of correlation time. The experimental data are then presented in a Table form, but I would suggest the authors indicate them by lines on the theoretical curves to allow easier interpretation by the reader.  The heteronuclear NOE values indicate the surprising finding that the position in the peptide rather than being a (4R)- or (4S)-proline dictates the dynamic behaviour ? This is puzzling, and so is the large exchange contribution to the R2 rates. The delay between pulses is 400μs, so this implies movements on the millisecond time scale ? Finally, for the MpRS peptide, do both lines for the (4R)-FP have similar relaxation parameters ?

They finally look into the binding to the SH3 domain by a titration experiment, and measure both a protein 1H-15N spectrum and a direct 19F spectrum. Both peptides interact, but with a threefold different affinity and apparently different mode. The amplitude of the 19F CSPs in the MpSR peptide are different from those in the MpRS peptide, even for the residue in position 4 that should not interact (lines 114-115) ? I would have expected the red spectrum in Figure 6C to be identical to the ones of the free peptides (Figure 4), is there a referencing issue ? Finally, can the authors distinguish the major and minor 19F signals in the interacting spectra, or is line broadening too important ?

In conclusion, this is a thorough study of the influence of a fluorinated proline in a peptide motif, that should lay the basis for further use of this residue in advanced protein studies.

Minor remarks

Line 126 strong $H_\delta$(i) to $H_\alpha$(i-1) NOE

Line 549 Acknowledgments

---

## Author Response (AR1)

Referee 1:

This is a very interesting paper that looks at the structural and dynamical aspects of the introduction of fluorine at the 4-position for a proline residue in a stereospecific fashion. Some of the results are quite unexpected and that suggests a level of caution that should be employed while introducing 4R- or 4S- fluoroprolines to quantitively probe protein interactions involving proline-rich segments. Conversely, conformational biases can be introduced to probe specific aspects of protein-protein interactions. Overall, the paper is well-written and well referenced; the analyses are robust and complete. This paper should be interest to the readership of Magnetic Resonance Discussions. I have a few minor suggestions and queries listed below:

We thank Ranajeet Ghose for his kind comments.

1. Table 1 lists the shifts for MpRS and MpSR peptides separately making comparison a bit cumbersome. I suggest that a two-column format that lists the corresponding shifts side by side be used.

We agree with the referee that a comparison of chemical shifts would be facilitated by a side-by-side presentation of Table 1, but our attempts to reformat the table systematically led to a loss in readability. Instead, we now provide a supplementary figure 2 that displays the comparison of proton and carbon chemical shifts at positions delta and alpha between the two peptides. These positions are relevant to assess possible changes in the structure and/or dynamics of the polyproline peptide.

2. I think on line 148 the authors mean $^3J_{F-\delta2}$ that shows a 5 Hz difference from the free amino acid.

We thank the referee for pointing out this mistake that has been corrected.

3. For Table 3, by the anti-symmetric component of the shift tensor, I assume that the authors mean the rank-1 component. Best to clarify that since this is generally neglected in most relaxation analyses.

Yes, that is correct. The first rank component is here effectively included in the relaxation analysis.

4. Line 325 appears to have a typo – it should read "the higher affinity for MpSR relative to MpRS."

This has been corrected.

5. For the $K_d$ calculations, while I agree that a combined analysis of fluorine and $^1H/^{15}N$ data is the most robust way to proceed, given that the affinity of the non-fluorinated peptide was determined using $^1H/^{15}N$ data only, it is worth also reporting just that analysis for the fluorinated peptides for completeness. If possible, I would also suggest a bulk measurement using ITC perhaps, given the somewhat strange behavior of the RS peptide. Though I admit that similar non-canonical binding models may complicate the ITC analysis.

The Kd values measured from the sole $^1H/^{15}N$ data for the two fluorinated peptides as

well as for the equivalent non-fluorinated peptide are now provided in supplementary Table 1.

While we agree with the referee that ITC data could provide interesting additional insights on the binding mechanism (by comparing relative enthalpic and entropic contributions), we rather restricted the scope of this manuscript to the information provided by an extensive analysis of the fluorine signal of fluoroprolines to show the potential of such analysis. We agree that further analysis remains to be conducted to reveal some aspects of the recognition of polyproline motifs by SH3 domains that have been overlooked until now.

Referee 2:

1. The authors present a detailed NMR investigation of two SH3 class II binding peptides, containing each a (4R)- and (4S)-fluoroproline but at different location. They first present the NMR assignment of the peptides, and exploit a high resolution NOESY-HSQC spectrum to assign all proline resonances.
In this aspect, I am suprised why they only give a single value for the Hδ protons, knowing that the two Hδ's can be distinguished, and actually give infomation on the flexibility of the residues (see for example Ahuja et al. JMB 2016).

We thank Guy Lippens for pointing this point to our attention. Both Hδ chemical shifts are provided in the table enabling the comparison of their differences to be performed as suggested in the reference. Beside P1 and fluorinated prolines, the profile of these chemical shift differences is very similar suggesting a similar local dynamics. This profile is shown in the supplementary figure 2 and a sentence has been added in the text to mention this with a link to the suggested reference:
" The dynamics of the non-fluorinated prolines are also not impacted by the insertion of either (4S)- or (4R)-FPro, as measured from the difference between the diastereotopic Hδ chemical shifts (Ahuja et al., 2016) (Supplementary Fig.3).

I also wonder whether the larger dispersion of the Pro Cδ carbons in the MpRS peptide compared to that of the MpSR peptide (Figure 2) has a meaning ? When I compare the Cδ spread of the two prolines flanking the 4R-FP in the MpRS peptide ($\Delta C\delta$(3-5) = -0.3ppm) with the same value in the MpSR peptide ($\Delta C\delta$(7-9) = +0.27ppm), I again wonder whether the chemical shift contains structural information.

The difference in chemical shift dispersion observed in the two peptides is indeed striking and has not escaped our attention, it is explicitly mentioned in line 180. We refer to a possible change of the psi dihedral angle of the two prolines preceeding the FPro (3 and 7) that display the largest difference but we refrained to further any structural interpretation due to the lack of structural data on FPro containing polypeptides.

Not only the ring pucker but also the backbone conformation of the proline is influenced by the fluorine incorporation, with the (4S)-FP favoring the cis conformation. Here, I am somewhat confused. If the fluorine spectrum of the (4R)-FP4 in MpRS shows a major and minor peak in a 1:3 ratio, what do they represent ? A major trans form and a minor cis form of this floroproline? But these should then also show up in the 1H-13C HSQC spectra, and the cis form should be characterized by a Hα-H cross peak ? And is the situation different for the (4R)-FP8 in the MpSR peptide ? What about

the (4S)-FP in both peptides ? Are they in the cis conformation ? Elucidating these points seems important for the interested reader.

The analysis of NOEs measured in $D_2O$ unambiguously shows that Pro-(4S)-FPro and Pro-(4R)-FPro peptide bonds are in the trans conformation in both peptides as indicated by the similar intensities of the Hα(i-1) - Hδ(i) NOEs observed in the four cases and the lack of Hα(i-1)-Hα(i) cross peak. The significant amount of a minor peak observed for the (4R)-FP in the MpRS peptide is therefore not related to a local change of conformation.
By performing analysis of the sample used in NMR measurements, we detected that 35% peak for (4R)-FPro observed in [19]F NMR spectrum of MpRS peptide corresponds to by-product with distinct retention time in reverse-phase HPLC trace and mass increase of 14 Da. MS-MS analysis enabled to localize this modification to the Pro1 residue in the sequence. Hydrolytic and/or oxidative modification could take place over long time during NMR measurements. The manuscript has been updated to mention the origin of the peak heterogeneity.

In order to characterize the movements of the FP rings, they turn to relaxation measurements. These are not easy to interpret, knowing the multiple dipolar terms and the important csa contribution. With the help of Spinach simulations, they obtain reasonable estimates for the different rates as a function of correlation time. The experimental data are then presented in a Table form, but I would suggest the authors indicate them by lines on the theoretical curves to allow easier interpretation by the reader.

Figure 5 has been modified to display the range of experimental relaxation parameters.

The heteronuclear NOE values indicate the surprising finding that the position in the peptide rather than being a (4R)- or (4S)-proline dictates the dynamic behaviour ? This is puzzling, and so is the large exchange contribution to the R2 rates. The delay between pulses is 400μs, so this implies movements on the millisecond time scale ? Finally, for the MpRS peptide, do both lines for the (4R)-FP have similar relaxation parameters ?

The reviewer is correct: the heteronuclear NOE points to a different dynamical behaviour for both segments of the peptide, irrespective of the FPro identity. Also ms time scale dynamics are present, mostly on the second polyproline stretch. We speculate that the specific flanking amino acid sequences, which differ for both polyproline stretches, may have an impact on the conformational ensemble of the polyproline segments, in a similar way as has been observed recently for other homopolymeric sequences. More investigation is required to investigate this. We have updated the text to mention the time scale and comment on its possible origin:
" This revealed about double values throughout, revealing exchange contributions on the ms time scale at both sites for both MpRS and MpSR peptides. As residual exchange contributions cannot be excluded in the CPMG experiment, an interpretation of transverse relaxation rates would also be unreliable. The origin of the exchange contribution is unclear, but possibly may arise from transient interactions between the polyproline segment and the flanking sequence (RVYK). Further studies will be required to investigate this unexpected finding."

The minor form of the (4R)-FPro residue in MpRS turns out to be an unexpected

impurity from the synthesis. The longitudinal relaxation parameters of this signal (R1: 2.18 s-1, rho: 1.95 s-1 and sigma: -0.13 s-1) are very similar to the major form (R1: 2.23 s-1, rho: 1.85 s-1 and sigma: -0.12 s-1). The transverse relaxation measured with the CPMG experiment is however significantly different (6.65 s-1 versus 8.5 s-1).

They finally look into the binding to the SH3 domain by a titration experiment, and measure both a protein 1H-15N spectrum and a direct 19F spectrum. Both peptides interact, but with a threefold different affinity and apparently different mode. The amplitude of the 19F CSPs in the MpSR peptide are different from those in the MpRS peptide, even for the residue in position 4 that should not interact (lines 114-115) ? I would have expected the red spectrum in Figure 6C to be identical to the ones of the free peptides (Figure 4), is there a referencing issue ? Finally, can the authors distinguish the major and minor 19F signals in the interacting spectra, or is line broadening too important ?

The observation of comparable CSP at position 4 and 8 was indeed unexpected and contradicts our initial expectations when the peptide were designed according the notion of Small Linear Motifs introduced by Toby Gibson (Tompa P, Davey NE, Gibson TJ, Babu MM (2014). Mol Cell 55(2):161–169.). In lines 114-115, we present the rational of the peptide design "position 4 falls outside the expected PXXPX+ binding motif". We also provide now a 3D model (Supplementary figure 1) of the complex to visualize the relative positions of the two fluorine atoms within the peptide. These comparable CSP may be either due to a specific geometry of the two polyproline segments induced by the serine residue that may bend the PPII helix positioning FP 4 close to the SH3 surface and/or a dynamic averaging of CSP values due to one-dimensional diffusion of the SH3 on the peptide. This interpretation has been added to the text.

It is true that the frequencies of fluorine lines measured for the final titration point do not match those measured for the free peptides (in figure 4). This is mainly due to the fact that the solvent conditions are different, while the spectra of free peptides were recorded in water, the titration was performed in a buffer suitable for the SH3 domain (40 mM phosphate pH 7). The measurement conditions are now explicitly mentioned in the legend of figure 4.
The line broadening of the minor forms are indeed to large preventing their specific measurement during titration experiments.

In conclusion, this is a thorough study of the influence of a fluorinated proline in a peptide motif, that should lay the basis for further use of this residue in advanced protein studies.

Minor remarks
Line 126 strong Hd(i) to Ha(i-1) NOE Line 549 Acknowledgments

The modification has been implemented and we thank Guy Lippens for his careful reading and discussion.

Referee 3

In the manuscript 'Fluorine NMR study of proline-rich sequences using fluoroprolines', Sinnaeve et al. develop 19F NMR of (4R) and (4S)-fluoroproline as a tool to probe the

conformation of proline residues within polyproline tracts, and the binding of such sequences to an SH3 domain. As noted by the authors, 19F NMR of fluoroproline is perhaps surprisingly undeveloped – although there are some precedents in the literature that have not been cited. As such, this work is a welcome contribution. However, it would be helpful **to be more explicit about the conclusions the authors actually draw from the present study**, beyond a mere demonstration of NMR prowess. **The analysis of SH3 binding is also weak,** and while additional experimental data may not be required, a more rigorous analysis of experimental uncertainties should be carried out. Lastly, the terminology of R/S fluoroprolines within RS/SR peptides, while logical, is certainly prone to confusion (at least for this reviewer!), and would benefit from a consistent colour coding and presentation within figures.

The conclusion has been modified to highlight the specific results brought by our study on the interaction between the Vinexin b SH3 and proline rich binding motifs as well as to better explicit the relevance of such approach for the study of larger interaction networks involving SH3 domains.

We have revised the analysis of the SH3 binding according to the specific comments made by the referee, in particular concerning the analysis of the uncertainties (see below).

We understand that the MpRS and MpSR terminology may be confusing. It highlights the subtle stereoisomeric difference between the two peptides and provides a non-ambiguous designation of the peptides. Thanks to the referee's comment, we believe that the source of confusion has been reduced by checking the consistency of the color coding throughout the manuscript. The figure 7 was indeed particularly confusing in the first version of the manuscript and has been revised.

Major points:

1. The authors state that 19F NMR of fluoroproline has not been explored. This is not completely accurate: Thomas et al. (2009, Chem Comm) report NMR of cis/trans isomerisation Ac-FPro-OMe, albeit without any apparent followup to larger peptides or proteins. Torbeev (2013), cited in the current paper, also present 19F NMR studies of cis/trans isomerisation of F2Pro-labelled ß2-microglobulin (e.g. Fig. S12). This was just the result of a very brief literature search, and may not be a complete list: the authors themselves should conduct a more careful survey and acknowledge prior work more fully.

We were familiar with the works discovered by the referee. Our statement aimed at pointing out the limited use of fluoroprolines in large peptides and proteins, not in single amino-acid molecules (such as Thomas et al.): "Surprisingly, despite the well-established use of FPro residues in chemical biology, they have so far not found any application as 19F NMR reporters in protein studies,.. ". With this statement, we were aiming for similar applications such as those known for the widely used fluorinated aromatic amino acids, such as interaction studies, or dynamics. We note that a similar conclusion was drawn recently in a review paper on fluorinated prolines (Verhoork et al., "Fluorinated Prolines as Conformational Tools and Reporters for Peptide and Protein Chemistry", Biochemistry, 2018). However, it is true that there are studies that have used 19F NMR simply to confirm the individual conformational state of the residue, such as Torbeev et al. We are also aware of one paper (Dietz et al.,

ChemBioChem, 2015) that used 19F NMR to monitor the folding/unfolding of a foldon peptide. To avoid misinterpretation, we have modified the statement to:

" Surprisingly, despite the well-established use of FPro residues in chemical biology, they have so far found very limited at-tention as 19F NMR reporters in protein studies, in contrast to aromatic amino acids (Verhoork et al., 2018). In the limited protein or peptide studies that have used 19F NMR, it was mainly used to confirm the local conformational state of the fluoroproline residue (Torbeev et al., 2013, Verhoork et al., 2018). To the best of our knowledge, only one study went fur-ther and exploited 19F NMR of a foldon domain peptide containing (4R)-FPro and (4S)-FPro residues to monitor the fold-ing/unfolding process as a function of temperature (Dietz et al., 2015)."

2. Figures: A consistent colour coding to distinguish R/S within RS/SR peptides would be extremely helpful. The relative left/right placement is also inconsistent, e.g. Fig. 1A/B vs Fig. 1D, Fig. 6A/B vs Fig. 7B/C, etc.

The figures 1D has been revised to match a consistent placement of 4R and 4S fluoroprolines respectively at the left and right sides. We added (4R)- and (4S)-FPro labels to figure 6 to ease its reading and figure 7 has been changed for consistent placement of RS and SR peptides in the left and right sides, respectively.

3. 'Minor forms of prolines': minor peaks are discussed on many occasions, and are attributed to cis/trans isomerisation of neighbouring residues: what is the evidence for this? Can impurities be ruled out, e.g. have independent samples been prepared and compared?

Indeed, by checking the purity of MpRS sample that showed 35.5% of second form for 4R-FPro residue we identified it as a by-product (with distinct retention time and mass increase of 14 Da). The text and legend to figure 4 have been updated:

"By analyzing this particular sample by analytical HPLC and mass-spectrometry we identified this species as an impurity (with mass increase of 14 Da that is localized to Pro1 residue based on tandem MS2 experiment). Other minor peaks can correspond to minor forms of the peptide where a single proline or fluoroproline is in the cis-form. Thus, additional peaks could be conformers and impurities."

4. P. 7, l. 156-158: I have no idea what the authors mean by 'dynamic frustration', but it seems like a very bold statement that should be explained and justified. As far as I can see, the authors simply observe that the endo/exo equilibrium is (a) different between 4R and 4S FPro, (b) unchanged within a polyproline peptide, and (c) has no effect upon the broader conformation of the peptide. Is this a fair summary? A more straightforward statement of conclusions would in general be welcome throughout this manuscript.

This summary is correct, although the conformational endo-exo biais of (4R)- and (4S)-fluoroprolines was known before and does not result from our study. The modified conclusion is now including our conclusions on the conformational preferences of FPro within a polyproline peptide.

Dynamic frustration can be defined as non-native dynamics resulting from distinct equilibrium dihedral angles and non-native intramolecular contacts. It is a more

extended concept than conformational frustration and includes modified rates of interconversion between different conformations in the altered conformational ensemble. In our study, the introduction of different fluoroprolines results in an altered equilibrium of conformations compared to the wild-type peptide, resulting in different binding affinities to the SH3 domain.

5. P. 9, l. 173: provide a reference for the PPII destabilising nature of 4S-FPro. Can the authors quantify the energetics of this a little more carefully, e.g. what is the expected effect on cis/trans equilibrium, and its impact on the stability of the peptide structure?

The effect of 4S-FPro on destabilizing the PPII conformation in collagen was studied by Raines and colleagues and destabilization due to incorporation of this residue was shown (e.g., Bretscher et al. J. Am. Chem. Soc. 2001, 123, 777 and Horng et al. Protein Science 2006 15, 74). Dissecting the energetic contributions of a single residue is more challenging, however and was not subject of the present study. In the mentioned sentence, we use the word "destabilizing" in qualitative rather than quantitative manner.

6. P. 11, l. 240-245: specify the CPMG frequency. Can the authors discuss the possible origins of the chemical exchange they identify?

The half-echo delay is now mentioned in the main text. We have added one sentence to discuss the possible origin of the exchange contribution:

" The origin of the exchange contribution is unclear, but possibly may arise from transient interactions between the polyproline segment and the flanking sequence (RVYK). Further studies will be required to investigate this unexpected finding."

7. P. 11, l. 254-258: After an extensive section of method development, the authors report two correlation times for FPro residues, but provide no interpretation or discussion of these results. What was the point of this measurement, and what is the significance of the result?

The likely cause of the different dynamical behavior is the differing nature of the flanking residues on both sides of the two homopolymer segments. Indeed, it has recently been shown for other homopolymers (notably polyglutamine) that the flanking regions greatly determine the conformational preference and dynamics. It is likely that something similar occurs here, though further investigation is required. We have added modified the text to add this point. Here, the intention was to demonstrate the potential of FPro residues and 19F to reveal this.

8. HSQC titration: provide results for the titration of non-fluorinated peptide. Provide concentrations/equivalents for data shown in Fig. 6 – assuming that the same concentrations are used for MpSR and MpRS titrations, the titration data look extremely similar, which is hard to reconcile with the reported three-fold difference in affinity. Provide axes for the inset figure panel. What HSQC pulse sequence was used for acquisition? Provide a table or plot comparing chemical shift perturbations between all three peptides and, if available, illustrate this on the structure of the SH3 domain. Where is Trp37 relative to the peptide binding site and expected location of the FPro residue? Trp37 is called a 'striking difference' but in reality the difference in bound chemical shifts appears to be extremely small.

- We added a supplementary figure 4 to show the fit with the non-fluorinated peptide.

- A factor of three in the binding strength is indeed not dramatically affecting a titration curve from the point of view of the receptor (the titrated molecule) while a stronger difference is observed for the titrating molecule. We provide now a Jupyter notebook showing that a three-fold difference in affinity has a great impact on the fluorine chemical shift evolution during the titration of the protein by the peptide while the impact on the protein chemical shifts is limited. This is due to the fact that the affinity has a pronounced effect on the peptide bound fraction at low concentrations of the peptide.

- The stoichiometry is now shown in Figure 7 and axes are shown for the inset of Figure 6.

- Details of the HSQC pulse sequence are now provided in the Material and method section.

- A supplementary table 2 is showing the chemical shift perturbation for the 3 peptides.

- The residues used for the binding study are shown on the 3D model built by homology modelling shown in Supplementary Figure 1. This figure clearly shows the implication of Trp37 in the polyproline binding site. The difference in the trajectories of Trp37 side-chain Hε-Nε resonance is indeed very small but consistently above the precision of the frequency measurements. We find that this is a notable difference, considering the known importance of this residue in the interaction and the remote location of the fluorine atom that excludes a direct effect of the fluorine atom on the tryptophane resonances. The supplementary figure 4 is showing the similarity between the Trp37 Hε-Nε cross peak observed for the titration with MpSR and the non-fluorinated peptide.

9. Have the authors considered 2D lineshape analysis of the HSQC titration to provide an independent assessment of binding kinetics? Can the authors comment on their decision to analyse 19F titration data in terms of CSPs and linewidths separately rather than directly via lineshape analysis, e.g. as performed by Stadmiller et al.?

We added a supplementary figure 5 to show that the lineshape of the fluorine signals are lorentzian. We therefore performed the analysis within the frame of a fast on and off binding kinetics. The challenge in interpreting the MpRS data was to find a unique model able to explain our observations prior the determination of the parameters of this model. This proved to be difficult due to the large number of possible models as compared to the modest number of observables we currently have.

10. 19F titration: how are spectra in Fig. 6C/D normalised? The authors claim there is 'strong' exchange broadening, but from the data presented this seems exaggerated. What software was used to fit linewidths, and how were minor peaks handled during this fitting? R2 rates should have units of s-1 not Hz, and uncertainties should be reported. The authors consider more complex binding mechanisms on the basis of these linewidth measurements, but from inspection of the signal-to-noise in the spectra of Fig. 6C/D I'm not sure that this is entirely justified. In any case, a more careful analysis of uncertainties would resolve this issue.

- The peaks intensities shown in Figure 6 Peak were normalized to account for the difference in peptide concentrations and number of scans used to record the spectrum. This is now specified in the legend of figure 6.

- We replaced 'strong' by 'significant' in the sentence. The reader can now get an insight on the quality of the line fits: we have added a supplementary figure 5 to show the goodness of fit for the first titration point, where the signal/noise ratio is the lowest. We also provide as supplementary material the Jupyter notebook that was used to fit the fluorine resonance lines. The units have been changed to s-1. The uncertainties in the R2 derived from the line widths are now shown in figure 7.

- While it is true that we initially considered more complex binding schemes, with two possible sites, all our analysis relies on a binding site: " Based on the goodness of fit reported as the reduced $\chi^2$, the ternary complex turned out to be unnecessary to explain the data, thus implying that only one SH3 binds to the peptide."

11. HSQC CSP analysis: I would suggest fitting (and plotting) data for individual residues and then averaging the results of the fit, rather than averaging the CSPs and performing a single fit. In performing a global analysis of HSQC and 19F CSPs, and 19F linewidths, how were uncertainties determined and the relative contributions of each measurement type weighted? Are fit results sensitive to this weighting?

- We have added a Supplementary table 1 that provides the dissociation equilibrium constants for every single residue that was considered for the average. As expected, we find a Kd value that is consistent with the one derived from fitting the sum of all composite chemical shifts.

- Incorporating fluorine chemical shifts into the fit led to a slight increase of the Kd value from 74 $\mu$M to 96 $\mu$M for the high affinity MpSR peptide. The rmsd on the fluorine and proton chemical shifts were 0.0027 and 0.0125 ppm for the fluorine and proton resonances, respectively while a rmsd value of 0.009 ppm was obtained when only the proton chemical shifts were considered. No scaling factor was used to weight the contributions of the fluorine and proton in the target function. The uncertainties derived from the sole fit are largely underestimated due to the uncertainty on the concentrations of interacting species, which may be difficult to assess. A reference carefully addressing this issue is provided (Koehler et al, Methods Mol. Biol. 1286 (2015) 279–296). This is why we report uncertainties on equilibrium constants that are larger than those resulting from the fit. Nevertheless, the factor of three between the two affinities is far above the measurement uncertainties and is highly significant. This is now better explained in the text.

12. What is the basis for relating the magnitude of 19F chemical shift perturbations to the strength of binding?

This is addressed in point 8 and in the associated Jupyter notebook that provides simulations of bound fractions of protein and ligand for different Kd.

[Figure]

13. Based on the extensive literature of SH3-peptide interactions, can the authors model the structure of the bound peptide, and perhaps examine the relative placement of 19F atoms and the intermolecular contacts that might be made?

Supplementary figure 1 shows a homology model of the complex together with the location of observed chemical shift perturbations.

14. In the discussion, the authors attempt to relate changes in 'conformational biases' with the effect on the binding equilibrium. However, their results indicate that the peptide conformations are in fact extremely similar - as gauged by near identical chemical shifts. The 'substantial shift' in binding affinity also corresponds to a very modest âˆ†âˆ†G of 0.6 kcal mol-1. In short, I struggle to understand the authors' interpretation of their results: more clarity is required.

We referred to the conformational bias of the ring conformation of fluorinated prolines. It is true that the effect of this local bias on the remaining part of the polyproline peptide is very limited, nevertheless it results into a weaker binding. In eukaryotic protein sequences, proline rich motifs are often clustered and minor changes in the binding affinity of a domain to a single site upon a chemical modification such as phosphorylation may lead to important biological effects if this change is correlated with other interactions. We believe that fluoroprolines represent an elegant tool to study such phenomenons. We have revised the conclusion to state more explicitly this point.

Minor points:

1. Fig. 2, caption: it would be helpful to note that this is the Cδ/Hδ region of the HSQC-NOESY rather than simply the Cδ region.

The legend of figure 2 has been revised accordingly:

"1H-13C HSQC-NOESY (mixing time: 80 ms) with a narrow 13C window focussing on the 13Cδ/Hδ correlations regions of both MpRS and MpSR peptides, recorded at 298 K and 700 MHz."

2. P. 1, l. 25: within
3. P. 9, l. 186: OMe

The two typos have been corrected.

4. Fig. 7: it's unclear what is being plotted in the RH panel of A. Chemical shifts should be provided in full on axes. What are the black/grey data in panel B? Are these fits to Eq. 1? The legend is unclear.

The figure 7 has been revised as well as the legend that presents explicitly the meaning of the solid lines in the panel C. These lines are now shown in black for both fluoroprolines to avoid confusion.

5. P. 16, l. 375: Hz or s-1?

kon rate constants are reported in M-1 S-1.

---

## Author Response (AR2)

Dear editor,

Thank you for taking care of editing our manuscript. We have deposited the NMR spectra on Zenodo and the corresponding archive is now properly referenced in the manuscript with a DOI. We did our best to format the manuscript according the guidelines recommended by Magnetic Resonance.

Yours,

Bruno Kieffer